# Sex differences in the sympathetic neurocirculatory responses to chemoreflex activation

Ana Luiza C. Sayegh[1] (ID), Jui-Lin Fan[1] (ID), Lauro C. Vianna[2] (ID), Mathew Dawes[3], Julian F. R. Paton[1] (ID) and James P. Fisher[1] (ID)

[1]*Manaaki Manawa – The Centre for Heart Research, Department of Physiology, Faculty of Medical and Health Sciences, University of Auckland, Auckland, New Zealand*
[2]*NeuroV˙ASQ˙ - Integrative Physiology Laboratory, Faculty of Physical Education, University of Brasília, Brasília, DF, Brazil*
[3]*Department of Medicine, Faculty of Medical and Health Sciences, University of Auckland, Auckland, New Zealand*

Edited by: Harold Schultz & Emma Hart

The peer review history is available in the supporting information section of this article (https://doi.org/10.1113/JP282327#support-information-section).

The Journal of Physiology

**Abstract** The purpose of this study was to determine whether there are sex differences in the cardio-respiratory and sympathetic neurocirculatory responses to central, peripheral, and combined central and peripheral chemoreflex activation. Ten women ($29 \pm 6$ years, $22.8 \pm 2.4$ kg/m$^2$: mean $\pm$ SD) and 10 men ($30 \pm 7$ years, $24.8 \pm 3.2$ kg/m$^2$) undertook randomized 5 min breathing trials of: room air (eucapnia), isocapnic hypoxia (10% oxygen ($O_2$); peripheral chemoreflex activation), hypercapnic hyperoxia (7% carbon dioxide ($CO_2$), 50% $O_2$; central chemoreflex activation) and hypercapnic hypoxia (7% $CO_2$, 10% $O_2$; central and peripheral chemoreflex activation). Control trials of isocapnic hyperoxia (peripheral chemoreflex inhibition) and hypocapnic hyperoxia (central and peripheral

**Ana Luiza Sayegh** completed her PhD at the University of Sao Paulo, Brazil, investigating the effectiveness of exercise rehabilitation in endomyocardial fibrosis. Following this she took up a postdoctoral fellowship in the Pulmonary Department at the Federal University of Sao Paulo where her work focused on the use of the cardiorespiratory responses to exercise on risk stratification and prognostication in pulmonary arterial hypertension. Currently, Ana is a postdoctoral fellow with Dr James Fisher in the Department of Physiology at the University of Auckland where she studies the neural control of the circulation in health and hypertension with a focus on the central and peripheral chemoreflexes.

chemoreflex inhibition) were also included. Muscle sympathetic nerve activity (MSNA; microneurography), mean arterial pressure (MAP; finger photoplethysmography) and minute ventilation ($\dot{V}_E$; pneumotachometer) were measured. Total MSNA ($P = 1.000$ and $P = 0.616$), MAP ($P = 0.265$) and $\dot{V}_E$ ($P = 0.587$ and $P = 0.472$) were not different in men and women during eucapnia and during isocapnic hypoxia. Women exhibited attenuated increases in $\dot{V}_E$ during hypercapnic hyperoxia ($27.3 \pm 6.3$ *vs.* $39.5 \pm 7.5$ l/min, $P < 0.0001$) and hypercapnic hypoxia ($40.9 \pm 9.1$ *vs.* $53.8 \pm 13.3$ l/min, $P < 0.0001$) compared with men. However, total MSNA responses were augmented in women (hypercapnic hyperoxia $378 \pm 215$ *vs.* $258 \pm 107\%$, $P = 0.017$; hypercapnic hypoxia $607 \pm 290$ *vs.* $362 \pm 268\%$, $P < 0.0001$). No sex differences in total MSNA, MAP or $\dot{V}_E$ were observed during isocapnic hyperoxia and hypocapnic hyperoxia. Our results indicate that young women have augmented sympathetic responses to central chemoreflex activation, which explains the augmented MSNA response to combined central and peripheral chemoreflex activation.

(Received 1 September 2021; accepted after revision 25 April 2022; first published online 28 April 2022)

**Corresponding author** James P. Fisher: Manaaki Manawa – The Centre for Heart Research, Department of Physiology, Faculty of Medical and Health Sciences, University of Auckland, 85 Park Road, Grafton, Auckland 1142, New Zealand. Email: jp.fisher@auckland.ac.nz

**Abstract figure legend** The sympathetic neurocirculatory and cardiorespiratory responses to chemoreflex activation were compared in healthy women and men. We show that, compared with young men, young women display augmented increases in muscle sympathetic nerve activity during both central chemoreflex activation (hypercapnic hyperoxia) and combined central and peripheral chemoreflex activation (hypercapnic hypoxia) but have attenuated increases in minute ventilation.

## Key points

- Sex differences in the control of breathing have been well studied, but whether there are differences in the sympathetic neurocirculatory responses to chemoreflex activation between healthy women and men is incompletely understood.
- We observed that, compared with young men, young women displayed augmented increases in muscle sympathetic nerve activity during both hypercapnic hyperoxia (central chemoreflex activation) and hypercapnic hypoxia (central and peripheral chemoreflex activation) but had attenuated increases in minute ventilation.
- In contrast, no sex differences were found in either muscle sympathetic nerve activity or minute ventilation responses to isocapnic hypoxia (peripheral chemoreceptor stimulation).
- Young women have blunted ventilator, but augmented sympathetic responses, to central (hypercapnic hyperoxia) and combined central and peripheral chemoreflex activation (hypercapnic hypoxia), compared with young men. The possible causative association between the reduced ventilation and heightened sympathetic responses in young women awaits validation.

## Introduction

The central and peripheral chemoreceptors are well recognized for their key role in the control of respiration (Dejours, 1962), but also a range of other homeostatic mechanisms, most notably cardiovascular and sympathetic regulation (Kara et al., 2003; Zera et al., 2019). Sex differences in the ventilatory responses to central, peripheral, and combined central and peripheral chemoreflex activation have been extensively examined using hypercapnic hyperoxia (Kunitomo et al., 1988), isocapnic hypoxia (Aitken et al., 1986; Kunitomo et al., 1988) and hypercapnic hypoxia (McCauley et al., 1988; Regensteiner et al., 1988), respectively. However, the extent to which there are sex differences in the sympathetic neurocirculatory responses to chemoreflex activation remains incompletely understood. It is important to address this because notable sex differences have been identified in the sympathetic regulation of blood pressure (BP) (Hart & Charkoudian, 2014), sympathetic reactivity to physiological stressors (e.g. exercise, heat, mental and orthostasis) (Barnes & Charkoudian, 2021), and the prevalence of hypertension and sleep-disordered breathing (e.g. sleep apnoea) (Redline et al., 1994; Young et al., 1993).

During hypoxia, peak increases in vasoconstrictor sympathetic nerve activity (MSNA) directed to the skeletal muscle vasculature are reported to be similar in young men and women (Jones et al., 1999; Miller, Cui et al., 2019). However, the MSNA response to a volitional breath-hold performed after breathing a hypercapnic and hypoxic gas mixture is greater in women during the early follicular (low hormone) menstrual cycle phase compared with men (Usselman et al., 2015). This augmented response may be attributable to an enhanced central chemoreflex sympatho-excitatory drive; however, to the authors' knowledge, the MSNA responses to hypercapnic normoxia have not been compared in young men and women. Alternatively, a more pronounced synergistic interaction to combined central and peripheral chemoreflex activation may explain the augmented MSNA response to hypercapnic hypoxia in young women. Indeed, combined central and peripheral chemoreflex stimulation has been reported to result in a MSNA response that is more than twofold greater than the sum of the individual responses to central and peripheral chemoreflex stimulation alone, in mixed groups of men and women (Jouett et al., 2015; Somers et al., 1989).

A full understanding of the integrated control of BP by the central and peripheral chemoreflexes involves consideration of both neural (i.e. BP to MSNA) and peripheral (i.e. MSNA to BP) components (Ogoh et al., 2009). Central interactions between baroreflex and chemoreflex function have been demonstrated (Heistad et al., 1975), while baroreflex control of MSNA is impaired as a consequence of the repeated hypoxaemia and hypercapnia experienced by individuals with obstructive sleep apnoea (Narkiewicz et al., 1998). Hypercapnia and hypoxia elicit a rightward resetting of the arterial baroreflex control of MSNA, while sensitivity is preserved (Halliwill & Minson, 2002; Steinback et al., 2009), but whether this remains the case during hypercapnic hypoxia remains to be determined. Given that the arterial baroreflex can restrain the MSNA response to chemoreflex activation (Heusser et al., 2020) and other physiological stressors (Scherrer et al., 1990), sex differences in the baroreflex control of MSNA may contribute to the differential sympathetic responses to chemoreflex activation in men and women (Usselman et al., 2015). The moment-to-moment bursting of the sympathetic nervous system is also a key factor for the maintenance of BP. A MSNA burst elicits a transient increase in vascular resistance and BP, whereas in the absence of a MSNA burst, BP falls (Fairfax et al., 2013; Vianna et al., 2012; Wallin & Nerhed, 1982). Usselman et al. (2015) reported a tendency ($P = 0.06$) for the quotient of total peripheral resistance (TPR) and MSNA (i.e. used as an index of neurovascular transduction) to be blunted in women during a hypercapnic hypoxic apnoea compared with men. However, a comprehensive evaluation of the effects of hypercapnia, hypoxia and combined hypercapnic hypoxia has yet to be undertaken in the same groups of young men and women.

Herein, MSNA responses to isocapnic hypoxia (peripheral chemoreflex activation), hypercapnic hyperoxia (central chemoreflex activation) and hypercapnic hypoxia (combined central and peripheral chemoreflex activation) were determined in young men and women (i.e. primary outcome). Arterial baroreflex control of MSNA and sympathetic neurovascular transduction were calculated as secondary outcomes. Control trials of isocapnic hyperoxia (peripheral chemoreflex inhibition) and hypocapnic hyperoxia (central and peripheral chemoreflex inhibition) were also included. Given reports that peak sympatho-excitation is greater in young women (during the early follicular phase) than men during a hypercapnic hypoxic apnoea (Usselman et al., 2015), but not during isocapnic hypoxia (Jones et al., 1999; Miller, Cui et al., 2019), we hypothesized that the central chemoreflex control of MSNA is enhanced in young women.

## Methods

### Ethical approval

The study was approved by the Northern B Health and Disability Ethics Committee, Auckland, New Zealand (19/NTB/125) and registered in the Australian New Zealand Clinical Trials Registry (ACTRN12619001767190). Each participant signed a written consent form after being provided with a detailed verbal and written explanation of the study procedures. The study was conducted according to the *Declaration of Helsinki*.

### Participants

Ten women (age $29 \pm 5$ years, weight $64.2 \pm 9.1$ kg, height $1.68 \pm 0.8$ m, body mass index $22.8 \pm 2.4$ kg/m$^2$: mean $\pm$ SD) and 10 men ($30 \pm 7$ years, $78.5 \pm 12.7$ kg, $1.78 \pm 0.6$ m, $24.8 \pm 3.2$ kg/m$^2$; $P = 0.615$, $P = 0.010$, $P = 0.005$ and $P = 0.125$, respectively) were recruited. The participants expressed the following cultural affiliations: Middle Eastern/Latin American/African ($n = 4$, 2 women), Asian ($n = 5$, 2 women) and European ($n = 11$, 6 women). No participant had a history or symptoms of pulmonary, metabolic (e.g. diabetes) or neurological disease, nor any acute or chronic disorders associated with alterations in cardiovascular structure or function. All participants reported being non-smokers, not users of recreational drugs, not abusers of alcohol and were not taking any medication apart from oral contraceptives. Women were studied either during the first 5 days of their menstrual cycle (early follicular phase) ($n = 8$) or during the placebo/no-hormone phase of oral contraceptive

use ($n = 2$). Prior to the experimental sessions, participants were requested to abstain from caffeinated beverages and physical activity for $>12$ h, alcohol for $>24$ h and food for $>2$ h. All experiments were performed in a temperature-controlled room (21–22°C).

### Experimental measurements

Participants were instrumented for continuous monitoring of heart rate (HR) by a lead II electrocardiogram (BioAmp, FE231, ADInstruments, Bella Vista, NSW, Australia) and beat-to-beat arterial pressure via finger photoplesthysmography (Human NIBP Nano interface, MLA382, ADInstruments). Validation of the finger BP values was achieved using brachial artery BP measurements obtained using an automated digital sphygmomanometer (Tango M2 BP monitor, SunTech, Morrisville, USA). Arterial oxygen saturation ($SpO_2$) was monitored with a pulse oximeter (MLT321 and ML320/F, ADInstruments). Recordings of postganglionic multiunit MSNA were obtained using the microneurography technique (Fisher et al., 2015). Briefly, a unipolar tungsten microelectrode was placed into the peroneal nerve at the fibular head, and a reference electrode inserted subcutaneously 2–3 cm distal. The recording electrode was manipulated in order to acquire a signal displaying a pulse-synchronous pattern of spontaneous sympathetic bursting activity that had a signal-to-noise ratio of 3:1 and was augmented at the end of a breath-hold but was unresponsive to skin stroking or an unexpected loud noise. The raw signal was amplified ($\times\ 100{,}000$), band-pass filtered (700–2000 Hz), rectified and integrated (time constant 0.1 s) to obtain a mean voltage neurogram (Iowa Bioengineering, Iowa City, IA).

Participants wore a facemask (V2 series, Hans Rudolph Inc., Shawnee, KS, USA) connected in series to a pneumotachometer (3830 Series, Heated Linear E Pneumotachometer; Hans Rudolph Inc.) and a non-rebreathing two-way valve (T-shape manual directional control stopcock-type, Hans Rudolph Inc.). Minute ventilation ($\dot{V}_E$), tidal volume ($V_T$) and respiratory frequency (R$f$) were measured breath by breath with the pneumotachograph (3830 Series, Heated Linear E Pneumotachometer, Hans Rudolph Inc.). End-tidal oxygen ($P_{ET}O_2$) and carbon dioxide ($P_{ET}CO_2$) were monitored continuously at the mouth using a gas analyser (Respiratory Gas Analyzer, ML206, ADInstruments) calibrated using standard gases. The breathing circuit was switched from breathing room air (open circuit) to inspiring from a 3 l bag containing the experimental gas mixtures (detailed below) using a three-way valve (2100 Series, Hans Rudolph Inc.). The perception of breathlessness was evaluated using a Borg scale (Borg, 1982), where each participant rated the difficulty of their breathing on a scale ranging from 'Your breathing

is causing you no difficulty at all' (0 points) to 'Your breathing difficulty is maximal' (10 points).

### Experimental protocol

The experimental protocol involved two visits to the laboratory, conducted on separate days. The initial familiarization visit involved a careful introduction to all non-invasive study methods and a full run-through of the study protocol. Anthropometric (height and weight), demographic and general health information were collected at this visit.

Upon arrival at the laboratory for the experimental visit, participants were positioned in a semi-recumbent position on a medical examination table. After instrumentation, a 20 min baseline in room air was recorded (eucapnia; last 4 min used for analysis) and followed by chemoreflex testing. The breathing trials used were isocapnic hypoxia (10% $O_2$, 90% nitrogen ($N_2$)) to activate the peripheral chemoreflex, hypercapnic hyperoxia (7% $CO_2$, 50% $O_2$, 43% $N_2$) to activate the central chemoreflex while inhibiting the peripheral chemoreflex, and hypercapnic hypoxia (7% $CO_2$, 10% $O_2$, 83% $N_2$) to activate both the central and peripheral chemoreflexes. Two additional control trials were also undertaken: isocapnic hyperoxia (50% $O_2$, 50% $N_2$) to inhibit the peripheral chemoreflex, and hypocapnic hyperoxia (50% $O_2$, 50% $N_2$), to inhibit both the central and peripheral chemoreflexes. Isocapnic conditions were achieved by the titration of $CO_2$ into the inspirate as required to maintain $P_{ET}CO_2$ at eucapnic values. Hypocapnia was achieved by guiding the participants, with the aid of a metronome, to hyperventilate such that $P_{ET}CO_2$ was lowered by 5 mmHg from eucapnic values. Each trial lasted 5 min, was administered in a random order, and followed by $\geq 10$ min recovery until restoration of measured variables.

### Data analysis

Raw signals underwent analogue-to-digital conversion at 1 kHz (Powerlab and LabChart v8; ADInstruments) and were stored for offline analysis. HR was calculated on a beat-to-beat basis from the ECG. Beat-to-beat systolic (SBP) and diastolic (DBP) BP were obtained from the arterial BP waveform and mean arterial pressure (MAP) was obtained by integration of the arterial BP waveform over the entire cardiac cycle. Stroke volume (SV) was provided by integration of the computed aortic outflow waveform using the Modelflow method. Cardiac output (CO) was calculated as the product of HR and SV, while TPR was calculated as MAP/CO. The Mosteller formula (Lam & Leung, 1988) was used to estimate body surface area and to calculate the TPR index ($TPR_i$) and CO index ($CO_i$) (Usselman et al., 2015). The capnograph trace was

left-shifted relative to the haemodynamic variables time series by 2.72 s to account for the gas measurement delay due to the physical components of the sampling circuit (Peebles et al., 2012).

MSNA bursts were identified using an interactive Spike 2 scoring script (Cambridge Electronic Design, Cambridge, United Kingdom). To account for the sympathetic nerve conduction delay, neurograms were shifted in time according to participants' height (Fagius & Wallin, 1980). The neurogram baseline was established (i.e. zero), and the identification of the largest spontaneous burst, not occurring with any provocation or outside stimulation, was set at 100 arbitrary units (a.u.) and all other bursts were expressed in proportion to this value. Identified bursts were inspected and scored (burst or no burst) by a single operator (A.L.C.S.). MSNA was quantified as burst frequency (bursts per minute), burst incidence (number of bursts per 100 heart beats), burst amplitude (i.e. strength) and total activity (product of burst frequency and mean burst amplitude) (Fisher et al., 2018).

Arterial baroreflex control of MSNA (ABR-MSNA) was determined from the relationship between DBP *vs.* burst incidence and total MSNA (Shantsila et al., 2015; Sundlof & Wallin, 1978). Sympathetic neurovascular transduction was indexed first, by the quotient of $TPR_i$ and total MSNA following the approach of Usselman et al. (2015) and second, from the dynamic relationship between changes in DBP following a spontaneous MSNA burst using a signal-averaging technique (Vianna et al., 2012). Spontaneous cardiac baroreflex sensitivity (cBRS) was calculated using the sequence technique (Cardio-Series v2.7, Ribeirão Preto, SP, Brazil) (Fisher et al., 2009; Parati et al., 2000). Baroreflex effectiveness index (BEI) was calculated as the ratio between the total number of baroreflex sequences and the total number of SBP ramps, where a 'SBP ramp' was defined as a progressive increase or decrease in SBP occurring over $\geq 3$ consecutive beats (Di Rienzo et al., 2001). Heart rate variability (HRV) was assessed following the guidelines provided by the Task Force of the European Society of Cardiology and the North American Society of Pacing and Electrophysiology (Anonymous, 1996) using CardioSeries v2.7 software.

Cardiorespiratory and sympathetic variables were averaged over the last 4 min of each trial (including eucapnia), to avoid the period taken for $P_{ET}O_2$ and $P_{ET}CO_2$ to stabilize. Responses induced by isocapnic hypoxia, hypercapnic hyperoxia and hypercapnic hypoxia were quantified as the difference from eucapnia. The 'physiological sum' of the responses to isocapnic hypoxia (peripheral chemoreflex activation) and hypercapnic hyperoxia (central chemoreflex activation) was calculated for comparison with the response to hypercapnic hypoxia (combined peripheral and central chemoreflex sensitivity). This approach was used by Somers et al. (1989) to explore the nature of the interaction between the central and peripheral chemoreflexes. If the response to hypercapnic hypoxia (combined peripheral and central chemoreflex sensitivity) is not different to the sum of isocapnic hypoxia (peripheral chemoreflex activation) and hypercapnic hyperoxia (central chemoreflex activation), then it may be concluded that a simple algebraic summation of the central and peripheral chemoreflexes has occurred. However, if the response to hypercapnic hypoxia is greater than the sum of isocapnic hypoxia and hypercapnic hyperoxia, then it may be concluded that combined peripheral and central chemoreflex sensitivity evokes a hyper-additive response. Finally, if the sum is less than the summation of the component parts then an occlusive interaction may be concluded (Somers et al., 1989).

## Statistical analysis

Normality was assessed with the Shapiro–Wilk test. Demographic and anthropometric data for men and women were compared using Student's *t* test. The main effects of sex, breathing trial, and their interaction were examined using mixed model analysis of variance (ANOVA) with repeated measures. *Post hoc* analysis was undertaken using a *t* test with Bonferroni correction. $P < 0.05$ was considered significant. Values are presented as means $\pm$ SD, unless otherwise stated. Statistical analyses were performed using SigmaPlot version 14.0 (Systat Software, Inc., San Jose, CA, USA).

## Results

### Central and peripheral chemoreflexes: cardiorespiratory responses

Compared with eucapnia (room air), $SpO_2$ and $P_{ET}O_2$ were decreased during isocapnic hypoxia (peripheral chemoreflex activation) and hypercapnic hypoxia (combined central and peripheral chemoreflex activation) and increased during hypercapnic hyperoxia (central chemoreflex activation with peripheral chemoreflex inhibition) (Table 1, Figs 1 and 2). $P_{ET}CO_2$ was not different from eucapnia during isocapnic hypoxia but was increased during hypercapnic hyperoxia and hypercapnic hypoxia. $SpO_2$, $P_{ET}O_2$ and $P_{ET}CO_2$ were not different between women and men in any trial.

$\dot{V}_E$, $V_T$ and R*f* were not different in women and men during eucapnia (Table 1, Fig. 3). Hypercapnic hyperoxia and hypercapnic hypoxia increased $\dot{V}_E$, $V_T$ and R*f* in both women and men; however, men showed augmented $\dot{V}_E$ and $V_T$ responses. The physiological sum of the $\dot{V}_E$ responses to isocapnic hypoxia (women: $\Delta 2.9 \pm 3.5$ l min$^{-1}$ *vs.* men: $\Delta 3.4 \pm 4.2$ l min$^{-1}$) and

**Table 1. Cardiorespiratory and sympathetic variables during eucapnia, isocapnic hypoxia (peripheral chemoreflex activation), hypercapnic hyperoxia (central chemoreflex activation) and hypercapnic hypoxia (combined peripheral and central chemoreflex activation) in women and men**

| | | Eucapnia | Isocapnic hypoxia | Hypercapnic hyperoxia | Hypercapnic hypoxia | ANOVA $P$ values Sex | ANOVA $P$ values Trial | ANOVA $P$ values Interaction |
|---|---|---|---|---|---|---|---|---|
| **Respiration** | | | | | | | | |
| SpO$_2$ (%) | Women | 97 ± 1 | 86 ± 2 | 99 ± 1 | 89 ± 1 | 0.109 | <0.0001 | 0.065 |
| | Men | 97 ± 1 | 87 ± 3 | 99 ± 0 | 91 ± 2 | | a,b,c,d,e,f | |
| V$_T$ (l) | Women | 0.93 ± 0.27 | 0.92 ± 0.31 | 1.43 ± 0.41[†,‡] | 1.73 ± 0.41[†,‡,§] | 0.132 | <0.0001 | 0.006 |
| | Men | 0.85 ± 0.22 | 1.08 ± 0.30 | 1.81 ± 0.35[*,†,‡] | 2.06 ± 0.39[*,†,‡,§] | | | |
| R$f$ (breaths·min$^{-1}$) | Women | 13 ± 5 | 16 ± 5 | 18 ± 5 | 22 ± 5 | 0.517 | <0.0001 | 0.519 |
| | Men | 15 ± 4 | 16 ± 5 | 20 ± 4 | 24 ± 7 | | b,c,d,e,f | |
| Perception of breathlessness (a.u.) | Women | 0 ± 0 | 1 ± 1 | 5 ± 2 | 7 ± 2 | 0.699 | <0.0001 | 0.857 |
| | Men | 0 ± 0 | 1 ± 1 | 5 ± 1 | 7 ± 2 | | b,c,d,e,f | |
| **Cardiovascular** | | | | | | | | |
| HR (beats·min$^{-1}$) | Women | 70 ± 10 | 78 ± 10 | 73 ± 9 | 84 ± 12 | 0.706 | <0.0001 | 0.658 |
| | Men | 69 ± 7 | 75 ± 5 | 73 ± 6 | 83 ± 8 | | a,c,d,e,f | |
| SBP (mmHg) | Women | 113 ± 10 | 122 ± 9 | 125 ± 15 | 143 ± 17 | 0.011 | <0.0001 | 0.844 |
| | Men | 126 ± 10 | 134 ± 8 | 141 ± 14 | 153 ± 18 | | a,b,c,e,f | |
| DBP (mmHg) | Women | 71 ± 7 | 75 ± 5 | 79 ± 10 | 89 ± 9 | 0.861 | <0.0001 | 0.554 |
| | Men | 68 ± 12 | 75 ± 13 | 82 ± 14 | 88 ± 12 | | b,c,d,e,f | |
| MAP (mmHg) | Women | 85 ± 7 | 92 ± 5 | 96 ± 11 | 108 ± 11 | 0.265 | <0.0001 | 0.526 |
| | Men | 88 ± 12 | 98 ± 11 | 104 ± 15 | 112 ± 12 | | a,b,c,e,f | |
| CO (l·min$^{-1}$) | Women | 5.6 ± 1.7 | 6.2 ± 1.6 | 5.8 ± 1.6 | 7.0 ± 2.5 | 0.543 | <0.0001 | 0.216 |
| | Men | 5.9 ± 1.2 | 6.8 ± 1.5 | 6.6 ± 1.7 | 7.2 ± 1.8 | | a,c,e,f | |
| CO$_i$ (l·min$^{-1}$·m$^{-2}$) | Women | 3.3 ± 1.1 | 3.6 ± 1.0 | 3.4 ± 1.1 | 4.1 ± 1.5 | 0.676 | <0.0001 | 0.158 |
| | Men | 3.1 ± 0.7 | 3.5 ± 0.9 | 3.4 ± 1.0 | 3.7 ± 1.1 | | a,c,e,f | |
| SV (ml) | Women | 84 ± 20 | 82 ± 22 | 83 ± 21 | 86 ± 28 | 0.582 | 0.939 | 0.179 |
| | Men | 87 ± 18 | 90 ± 18 | 91 ± 23 | 86 ± 20 | | | |
| TPR (mmHg·l$^{-1}$·min$^{-1}$) | Women | 16.5 ± 5.4 | 16.2 ± 5.6 | 18.3 ± 8.0 | 17.6 ± 8.0 | 0.742 | 0.004 | 0.998 |
| | Men | 15.6 ± 4.5 | 15.4 ± 5.4 | 17.3 ± 7.1 | 16.7 ± 5.0 | | b,d | |
| TPR$_i$ (mmHg·l$^{-1}$·min$^{-1}$·m$^{-2}$) | Women | 9.5 ± 2.7 | 9.3 ± 2.7 | 10.5 ± 3.8 | 10.1 ± 3.9 | 0.229 | 0.004 | 0.992 |
| | Men | 7.9 ± 2.1 | 7.8 ± 2.5 | 8.8 ± 3.3 | 8.5 ± 2.2 | | b,d | |
| **Sympathetic** | | | | | | | | |
| MSNA BF (bursts·min$^{-1}$) | Women | 11 ± 3 | 15 ± 4 | 22 ± 6[†‡] | 30 ± 8[†‡§] | 0.468 | <0.0001 | 0.027 |
| | Men | 11 ± 5 | 15 ± 5 | 19 ± 6[†] | 24 ± 6[*,†,‡] | | | |
| MSNA BI (bursts·100 heartbeats$^{-1}$) | Women | 16 ± 2 | 19 ± 2 | 30 ± 3[†,‡] | 36 ± 3[†,‡] | 0.574 | <0.0001 | 0.015 |
| | Men | 17 ± 2 | 20 ± 2 | 25 ± 2[†,‡] | 30 ± 2[†,‡] | | | |
| MSNA amplitude (%) | Women | 100 ± 0 | 119 ± 35 | 195 ± 34[†,‡] | 218 ± 63[†,‡] | 0.104 | <0.0001 | 0.031 |
| | Men | 100 ± 0 | 131 ± 45 | 147 ± 56[*] | 170 ± 64[*,†] | | | |

All variables are $n = 10$ women and $n = 10$ men. The main effects of sex, breathing trial and their interaction were examined using mixed model ANOVA with repeated measures. SpO$_2$, oxygen saturation; V$_T$, tidal volume; R$f$, breathing frequency; a.u., arbitrary units; HR, heart rate; SBP, systolic blood pressure; DBP, diastolic blood pressure; MAP, mean arterial pressure; CO, cardiac output; CO$_i$, cardiac output index; SV, stroke volume; TPR, total peripheral resistance; TPR$_i$, total peripheral resistance index; MSNA, muscle sympathetic nerve activity; BF, burst frequency; BI, burst incidence. Where a significant interaction is observed, differences identified during *post hoc* analysis (*t* tests with Bonferroni correction) are identified as [*] $P < 0.05$ *vs.* women, [†] $P < 0.05$ *vs.* eucapnia, [‡] $P < 0.05$ *vs.* isocapnic hypoxia, [§] $P < 0.05$ *vs.* hypercapnic hyperoxia. Where a significant main effect of trial, but no interaction, is observed, differences identified during *post hoc* analysis (*t* tests with Bonferroni correction) are shown as [a] $P < 0.05$ eucapnia *vs.* isocapnic hypoxia, [b] $P < 0.05$ eucapnia *vs.* hypercapnic hyperoxia, [c] $P < 0.05$ eucapnia *vs.* hypercapnic hypoxia, [d] $P < 0.05$ isocapnic hypoxia *vs.* hypercapnic hyperoxia, [e] $P < 0.05$ isocapnic hypoxia *vs.* hypercapnic hypoxia, [f] $P < 0.05$ hypercapnic hyperoxia *vs.* hypercapnic hypoxia.

hypercapnic hyperoxia (women: $\Delta 15.2 \pm 6.7$ l min$^{-1}$ *vs.* men: $\Delta 25.6 \pm 6.7$ l min$^{-1}$) was lower ($P < 0.0001$) than the responses to hypercapnic hypoxia (women: $\Delta 28.8 \pm 9.3$ l min$^{-1}$ *vs.* men: $\Delta 39.9 \pm 12.4$ l min$^{-1}$) in both women and men, respectively, indicative of a hyper-additive response. The perception of breathlessness increased during hypercapnic hypoxia compared with hypercapnic hyperoxia ($P < 0.0001$), isocapnic hypoxia ($P < 0.0001$) and eucapnia ($P < 0.0001$). This perception was not different between men and women ($P = 0.699$) and increased during hypercapnic hyperoxia ($P < 0.0001$) and hypercapnic hypoxia ($P < 0.0001$) compared with eucapnia.

No sex differences in diastolic and mean BP, HR, CO, CO$_i$, TPR and TPR$_i$ were noted during eucapnia (Table 1). Systolic BP was lower ($P = 0.011$) in women during eucapnia, isocapnic hypoxia, hypercapnic hyperoxia and hypercapnic hypoxia compared with men. Compared with eucapnia, HR was similarly elevated during isocapnic hypoxia and hypercapnic hypoxia in women and men. Likewise, MAP was similarly increased in women and men during isocapnic hypoxia ($P < 0.0001$), hypercapnic hyperoxia ($P < 0.0001$) and hypercapnic hypoxia ($P < 0.0001$) compared with eucapnia. CO and CO$_i$ increased similarly in women and men during isocapnic hypoxic ($P = 0.002$) and hypercapnic hypoxia ($P < 0.0001$) compared with eucapnia, while TPR ($P = 0.028$) and TPR$_i$ increased only during hypercapnic hyperoxia ($P = 0.030$) compared with eucapnia, with no difference between women and men.

## Central and peripheral chemoreflexes: sympathetic responses

MSNA (burst frequency, burst incidence) was not different in women and men during eucapnia (Table 1, Fig. 3). MSNA burst frequency, burst incidence, burst amplitude and total activity were increased during hypercapnic hyperoxia and hypercapnic hypoxia in women and men compared with eucapnia (Table 1, Fig. 3). Responses were greater in women for MSNA burst frequency during hypercapnic hypoxia ($P = 0.034$), for MSNA amplitude during hypercapnic hyperoxia ($P = 0.018$) and hypercapnic hypoxia ($P = 0.017$), and for total MSNA during hypercapnic hyperoxia ($P = 0.017$) and hypercapnic hypoxia ($P < 0.0001$) compared with men. The physiological sum of the changes in total MSNA during isocapnic hypoxia (women: $\Delta 51 \pm 61\%$ *vs.* men: $\Delta 40 \pm 36\%$) and hypercapnic hyperoxia (women: $\Delta 278 \pm 215\%$ *vs.* men: $\Delta 158 \pm 107\%$), was lower than the responses to

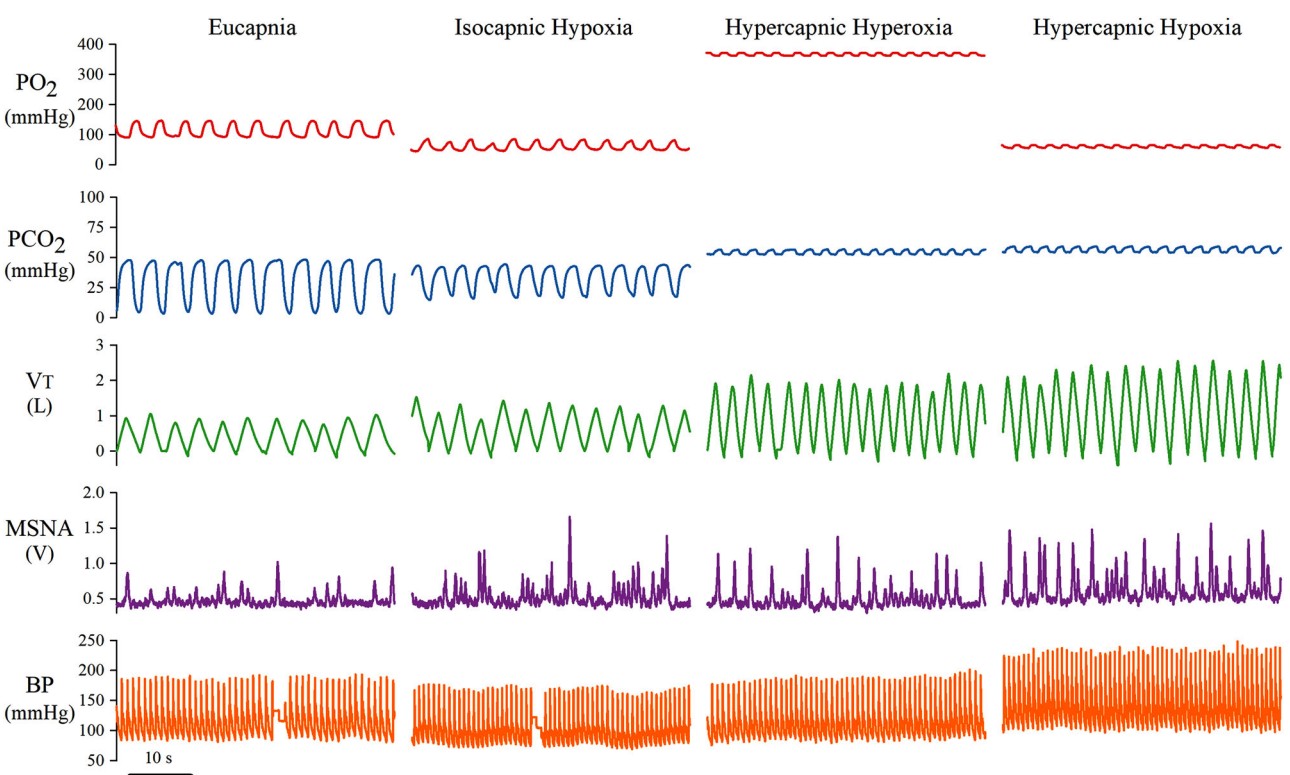

**Figure 1. Original records illustrating the cardiorespiratory and sympathetic responses to eucapnia, isocapnic hypoxia, hypercapnic hyperoxia and hypercapnic hypoxia in one individual**
PO$_2$, partial pressure of oxygen; PCO$_2$, partial pressure of carbon dioxide; V$_T$, tidal volume; MSNA, muscle sympathetic nerve activity; BP, blood pressure. [Colour figure can be viewed at wileyonlinelibrary.com]

hypercapnic hypoxia (women: $\Delta 507 \pm 290\%$ *vs.* men: $\Delta 262 \pm 268\%$) in both men and women ($P = 0.048$), respectively, indicative of a hyper-additive response (Fig. 3).

### Arterial baroreflex function and HRV

ABR-MSNA was not different between women and men during any breathing trial (Table 2, Fig. 4). ABR-MSNA, when expressed as total MSNA gain but not burst incidence, was increased during hypercapnic hyperoxia ($P = 0.012$) compared with eucapnia. cBRS gain was

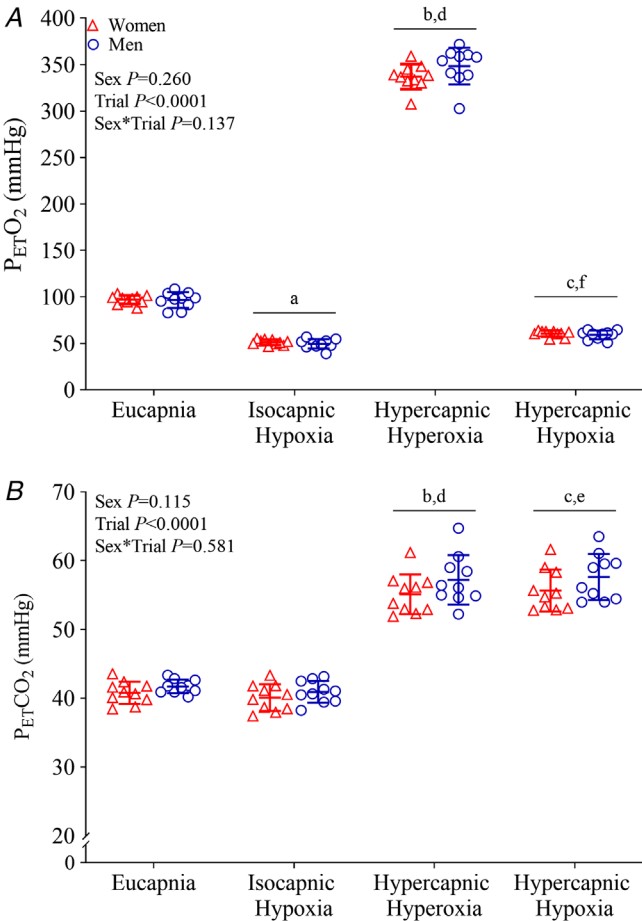

**Figure 2. Partial pressure of end-tidal oxygen (*A*; $P_{ET}O_2$) and carbon dioxide (*B*; $P_{ET}CO_2$) during eucapnia, isocapnic hypoxia, hypercapnic hyperoxia and hypercapnic hypoxia in women (red triangles, *n* = 10) and men (blue circles, *n* = 10)**
The main effects of sex, breathing trial and their interaction were examined using mixed model ANOVA with repeated measures. Where a significant main effect of trial, but no interaction, is observed, differences observed during *post hoc* analysis (*t* tests with Bonferroni correction) are shown as ${}^aP < 0.05$ eucapnia *vs.* isocapnic hypoxia, ${}^bP < 0.05$ eucapnia *vs.* hypercapnic hyperoxia, ${}^cP < 0.05$ eucapnia *vs.* hypercapnic hypoxia, ${}^dP < 0.05$ isocapnic hypoxia *vs.* hypercapnic hyperoxia, ${}^eP < 0.05$ isocapnic hypoxia *vs.* hypercapnic hypoxia, ${}^fP < 0.05$ hypercapnic hyperoxia *vs.* hypercapnic hypoxia. [Colour figure can be viewed at wileyonlinelibrary.com]

not different from eucapnia during any breathing trial in women and men but was increased during hypercapnic hyperoxia compared with hypercapnic hypoxia ($P = 0.046$). Women and men had fewer baroreflex sequences and a lower BEI during hypercapnic hyperoxia ($P = 0.014$ and $P = 0.046$, respectively) and hypercapnic hypoxia ($P = 0.002$ and $P < 0.0001$, respectively) compared with eucapnia. None of the HRV indices were different between women and men during any trial. However, the root mean square of successive differences (RMSSD), the standard deviation of normal sinus R–R intervals (SDNN), high frequency (HF) and total power were reduced during isocapnic hypoxia and hypercapnic hypoxia compared with eucapnia. Low frequency (LF) was lower during all trials *vs.* eucapnia.

### Sympathetic neurovascular transduction

Sympathetic neurovascular transduction, assessed from the quotient of $TPR_i$ and total MSNA, was similarly decreased during hypercapnic hyperoxia and hypercapnic hypoxia compared with eucapnia and isocapnic hypoxia, in women and men (Fig. 5). Sympathetic neurovascular transduction, assessed from the magnitude of the peak rise in DBP following multiple MSNA bursts, was blunted during isocapnic hypoxia ($P = 0.004$), hypercapnic hyperoxia ($P = 0.008$) and hypercapnic hypoxia ($P = 0.042$ *vs.* eucapnia) in women and men. However, for single bursts the magnitude of the peak DBP response was not different between trials in women and men (Fig. 5), and for all bursts the magnitude of the peak DBP response was attenuated only during hypercapnic hyperoxia ($P = 0.031$ *vs.* eucapnia; Table 3). Importantly, sympathetic neurovascular transduction was not different between men and women during eucapnia, isocapnic hypoxia, hypercapnic hyperoxia and hypercapnic hypoxia trials (Fig. 5, Table 3).

### Central and peripheral chemoreflexes: control trials

Cardiorespiratory and sympathetic responses to isocapnic hyperoxia (peripheral chemoreflex inhibition) and hypocapnic hyperoxia (central and peripheral chemoreflex inhibition) in women and men are provided in Table 4. By design, $SpO_2$ and $P_{ET}O_2$ were similarly elevated in women and men during isocapnic hyperoxia and hypocapnic hyperoxia compared with eucapnia, while $P_{ET}CO_2$ was reduced in hypocapnic hyperoxia ($P < 0.0001$ *vs.* eucapnia, and $P = 0.002$ *vs.* isocapnic hyperoxia). Overall, SBP was lower in women, while no other sex differences were evident. Both trials similarly increased MAP ($P < 0.0001$) and TPR ($P < 0.0001$ *vs.* eucapnia), while only hypocapnic hyperoxia increased $\dot{V}_E$, R*f* and DBP. Isocapnic hyperoxia increased MSNA burst amplitude ($P = 0.047$ *vs.* eucapnia) and HR ($P = 0.002$

*vs.* eucapnia and $P = 0.016$ *vs.* hypocapnic hyperoxia). Neither cBRS nor HRV (e.g. RMSSD, SDNN) was different between women and men during any trial. Baroreflex sequences ($P < 0.0001$) and BEI ($P = 0.001$ *vs.* eucapnia) were decreased during hypocapnic hyperoxia, as were SDNN ($P < 0.0001$), LF ($P = 0.007$) and total power ($P < 0.0001$ *vs.* eucapnia and isocapnic hyperoxia), while RMSSD ($P = 0.003$) and HF ($P = 0.011$) were decreased during hypocapnic hyperoxia compared with isocapnic hyperoxia (Table 5).

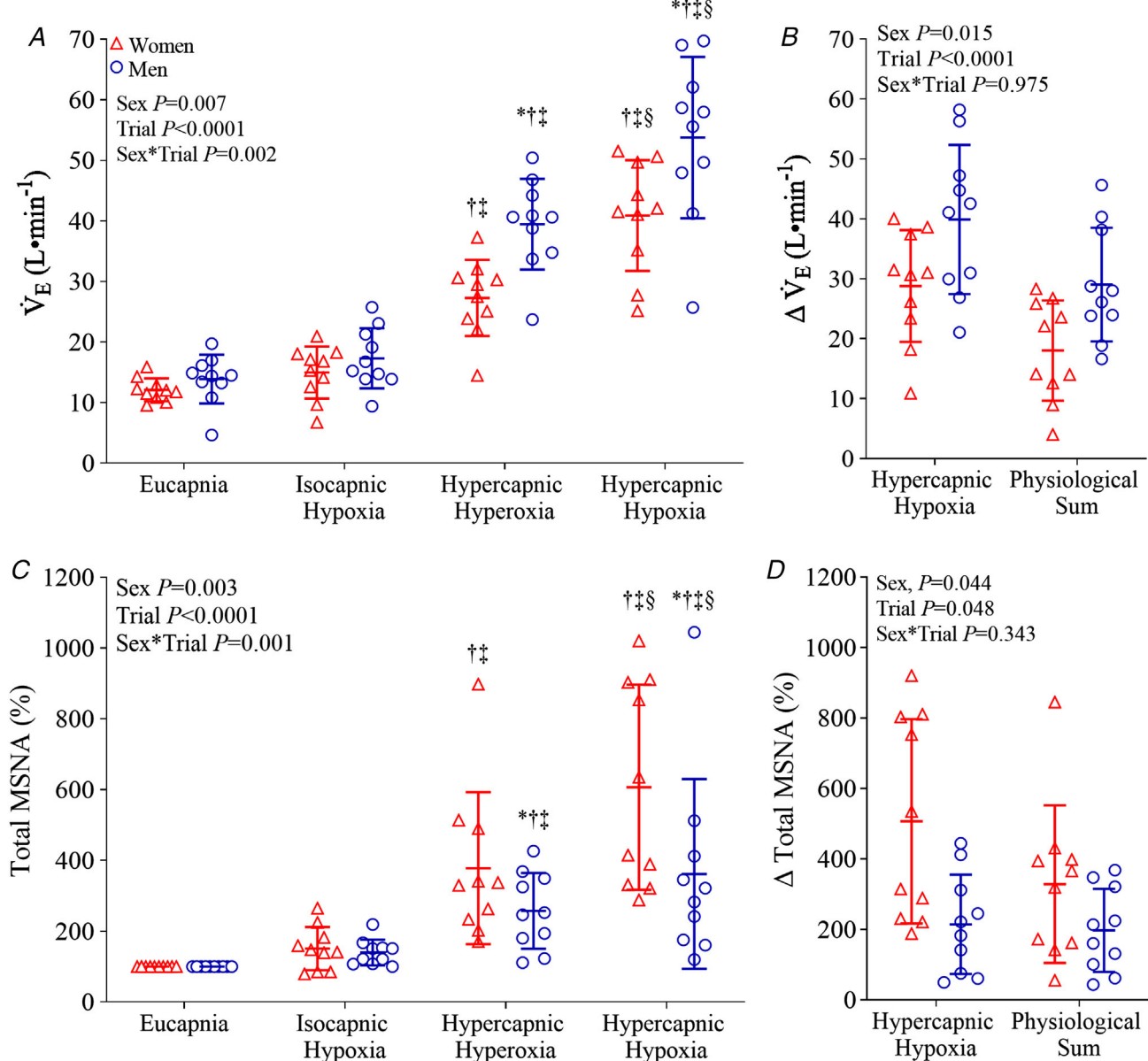

**Figure 3. Minute ventilation ($\dot{V}_E$) and muscle sympathetic nerve activity (MSNA; total activity) during eucapnia, isocapnic hypoxia, hypercapnic hyperoxia and hypercapnic hypoxia in women (red triangles, *n* = 10) and men (blue circles, *n* = 10)**
*A* and *C*, absolute values. *B* and *D*, comparison of the change with the combined hypercapnic hypoxia trial *vs.* the physiological sum of the responses to the separate isocapnic hypoxia and hypercapnic hyperoxia trials. The main effects of sex, breathing trial and their interaction were examined using mixed model ANOVA with repeated measures. Where a significant interaction is observed, differences identified during *post hoc* analysis (*t* tests with Bonferroni correction) are identified as *$P < 0.05$ *vs.* women, †$P < 0.05$ *vs.* eucapnia, ‡$P < 0.05$ *vs.* isocapnic hypoxia, §$P < 0.05$ *vs.* hypercapnic hyperoxia, #$P < 0.05$ *vs.* hypercapnic hypoxia. [Colour figure can be viewed at wileyonlinelibrary.com]

**Table 2. Baroreflex sensitivity and heart rate variability during eucapnia, isocapnic hypoxia, hypercapnic hyperoxia and hypercapnic hypoxia in women and men**

| | | Eucapnia | Isocapnic hypoxia | Hypercapnic hyperoxia | Hypercapnic hypoxia | ANOVA *P* values Sex | Trial | Interaction |
|---|---|---|---|---|---|---|---|---|
| ***ABR-MSNA*** | | | | | | | | |
| Burst incidence (bursts [100 heartbeats]$^{-1}$· mmHg$^{-1}$) | Women | $-4.14 \pm 1.45$ | $-3.25 \pm 1.20$ | $-4.05 \pm 1.90$ | $-3.24 \pm 2.11$ | 0.903 | 0.076 | 0.881 |
| | Men | $-3.80 \pm 1.54$ | $-3.47 \pm 1.59$ | $-4.51 \pm 1.84$ | $-3.35 \pm 2.01$ | | | |
| Total MSNA (a.u.·beat$^{-1}$· mmHg$^{-1}$) | Women | $-2.50 \pm 0.81$ | $-2.45 \pm 1.01$ | $-4.42 \pm 2.34$ | $-3.35 \pm 2.07$ | 0.525 | 0.010 b,d | 0.758 |
| | Men | $-2.21 \pm 1.00$ | $-2.67 \pm 1.15$ | $-3.61 \pm 2.19$ | $-3.24 \pm 1.49$ | | | |
| ***cBRS*** | | | | | | | | |
| Gain (ms·mmHg$^{-1}$) | Women | $15.6 \pm 5.0$ | $10.7 \pm 4.5$ | $14.4 \pm 6.1$ | $12.0 \pm 6.0$ | 0.763 | 0.045 f | 0.211 |
| | Men | $14.2 \pm 5.2$ | $14.7 \pm 14.9$ | $19.5 \pm 10.1$ | $14.1 \pm 10.2$ | | | |
| Number of sequences (*n*) | Women | $33 \pm 20$ | $33 \pm 18$ | $18 \pm 17$ | $25 \pm 18$ | 0.154 | <0.0001 b,c,d,e | 0.861 |
| | Men | $23 \pm 17$ | $21 \pm 18$ | $10 \pm 7$ | $8 \pm 11$ | | | |
| BEI (%) | Women | $0.57 \pm 0.32$ | $0.53 \pm 0.26$ | $0.35 \pm 0.25$ | $0.26 \pm 0.15$ | 0.503 | <0.0001 b,c,e | 0.738 |
| | Men | $0.50 \pm 0.25$ | $0.38 \pm 0.33$ | $0.35 \pm 0.28$ | $0.19 \pm 0.11$ | | | |
| ***HRV*** | | | | | | | | |
| RMSSD (ms) | Women | $73.1 \pm 41.2$ | $39.1 \pm 15.8$ | $82.5 \pm 39.3$ | $51.0 \pm 31.0$ | 0.744 | <0.0001 a,c,d,f | 0.332 |
| | Men | $69.2 \pm 34.9$ | $51.5 \pm 24.3$ | $69.1 \pm 44.3$ | $40.6 \pm 25.7$ | | | |
| SDNN (ms) | Women | $77.8 \pm 31.4$ | $54.5 \pm 21.0$ | $76.4 \pm 27.6$ | $58.6 \pm 21.5$ | 0.549 | <0.0001 a,c,f | 0.198 |
| | Men | $76.2 \pm 31.3$ | $60.7 \pm 21.1$ | $62.1 \pm 23.6$ | $46.1 \pm 19.8$ | | | |
| HF (ms$^2$) | Women | $3223 \pm 4205$ | $703 \pm 456$ | $3457 \pm 2867$ | $1425 \pm 1761$ | 0.403 | 0.009 a,c | 0. 256 |
| | Men | $2560 \pm 2170$ | $1653 \pm 1739$ | $1785 \pm 1520$ | $741 \pm 848$ | | | |
| LF (ms$^2$) | Women | $2159 \pm 1948$ | $790 \pm 1520$ | $866 \pm 1364$ | $480 \pm 772$ | 0.986 | <0.0001 a,b,c | 0.917 |
| | Men | $2057 \pm 1909$ | $1061 \pm 1334$ | $709 \pm 497$ | $443 \pm 522$ | | | |
| Total power | Women | $4874 \pm 4908$ | $1251 \pm 1769$ | $3014 \pm 3979$ | $1175 \pm 2394$ | 0.738 | <0.0001 a,c | 0.178 |
| | Men | $4674 \pm 3971$ | $2115 \pm 4072$ | $2335 \pm 1840$ | $990 \pm 1353$ | | | |
| HF (n.u.) | Women | $60 \pm 27$ | $59 \pm 19$ | $77 \pm 16$ | $62 \pm 21$ | 0.403 | 0.061 | 0.589 |
| | Men | $58 \pm 22$ | $61 \pm 21$ | $66 \pm 18$ | $52 \pm 22$ | | | |
| LF (n.u.) | Women | $40 \pm 27$ | $41 \pm 19$ | $23 \pm 16$ | $38 \pm 21$ | 0.403 | 0.061 | 0.589 |
| | Men | $42 \pm 22$ | $39 \pm 21$ | $34 \pm 18$ | $48 \pm 22$ | | | |
| LF/HF ratio | Women | $2.62 \pm 4.78$ | $1.51 \pm 1.96$ | $0.49 \pm 0.59$ | $1.40 \pm 1.69$ | 0.923 | 0.209 | 0.681 |
| | Men | $1.67 \pm 2.35$ | $1.22 \pm 1.53$ | $0.85 \pm 0.95$ | $2.06 \pm 1.89$ | | | |

All variables are $n = 10$ women and $n = 10$ men, aside from ABR-MSNA where $n = 9$ women and $n = 9$ men (due to slopes $R^2 < 0.45$) and cBRS where $n = 7$ women and $n = 8$ men (due to an absence of sequences). The main effects of sex, breathing trial and their interaction were examined using mixed model ANOVA with repeated measures. ABR-MSNA, arterial baroreflex control of MSNA; MSNA, muscle sympathetic nerve activity; cBRS, spontaneous cardiac baroreflex sensitivity; BEI, baroreflex effectiveness index; HRV, heart rate variability; RMSSD, square root of the mean of the sum of successive differences in R–R interval; SDNN, standard deviation of all normal sinus R–R intervals; HF, high frequency; n.u., normalized units; LF, low frequency. Where a significant main effect of trial, but no interaction, is observed, differences identified during *post hoc* analysis (*t* tests with Bonferroni correction) are shown as [a]$P < 0.05$ eucapnia *vs.* isocapnic hypoxia, [b]$P < 0.05$ eucapnia *vs.* hypercapnic hyperoxia, [c]$P < 0.05$ eucapnia *vs.* hypercapnic hypoxia, [d]$P < 0.05$ isocapnic hypoxia *vs.* hypercapnic hyperoxia, [e]$P < 0.05$ isocapnic hypoxia *vs.* hypercapnic hypoxia, [f]$P < 0.05$ hypercapnic hyperoxia *vs.* hypercapnic hypoxia.

## Discussion

The aim of this study was to determine whether there are sex differences in the sympathetic neurocirculatory responses to central, peripheral, and combined central and peripheral chemoreflex activation. We observed that during hypercapnic hypoxia (central and peripheral chemoreflex activation) young women have an augmented increase in total MSNA (i.e. primary outcome), but a lower $\dot{V}_E$ increase. We show that this is attributable to an augmented total MSNA response, but a lower $\dot{V}_E$ increase, to central chemoreflex activation (hypercapnic hyperoxia) in women. The increase in MSNA and $\dot{V}_E$ during hypercapnic hypoxia was greater than the sum of the individual responses to isocapnic hypoxia and hyperoxic hypercapnia in both women and men. Importantly, this hyper-additive response between central and peripheral chemoreflex-mediated sympathetic activation was not different in men and women.

## Central and peripheral chemoreflexes

In contrast, to the control of breathing (Behan & Wenninger, 2008), few studies have investigated the sex differences in chemoreflex control of MSNA. In agreement with Miller, Cui et al. (2019), we observed no difference in MSNA between women and men during isocapnic hypoxia. While Jones et al. (1999) found no difference in the peak total MSNA response to 10 min isocapnic hypoxia, a shorter latency of response was observed in women such that total MSNA was lower in women during the first 5 min. Providing a definitive explanation for the discrepancy with our findings is difficult; however, the lower MSNA in the women compared with men reported by Jones et al. (1999) (18 *vs.* 24 bursts/min) but no difference in the current study, along with differences in the number of women studied (*n* = 7 *vs.* 10) and control of menstrual cycle phase (uncontrolled *vs.* early follicular phase) may be contributing factors.

In contrast to isocapnic hypoxia, we observed a greater MSNA response in women during hypercapnic hypoxia. This finding concurs with Usselman et al. (2015), who reported that the MSNA response to an apnoea performed following the breathing of a hypercapnic hypoxic gas mixture, was augmented in women during the early follicular (low hormone) menstrual cycle phase compared with men. We reasoned that the augmented response to hypercapnic hypoxia in women may be attributable to a greater central chemoreflex sensitivity and manifest as an augmented response to hypercapnic hyperoxia. In accordance with our hypothesis, women exhibited an enhanced MSNA response to hypercapnic hyperoxia (i.e. augmented central chemoreflex control of MSNA).

The effect of peripheral chemoreceptor activation on the $\dot{V}_E$ response to central chemoreceptor activation remains contentious. Animal studies have reported a negative or an additive interaction between peripheral and central chemoreceptors (Adams & Severns, 1982; Day & Wilson, 2009), while others reported that chemoreceptors showed a hyper-additive interaction and modulate the responsiveness of one another (Blain et al., 2010; Fitzgerald & Parks, 1971). In humans, combined peripheral and central chemoreflex stimulation resulted in an additive HR and BP response, while $\dot{V}_E$ and MSNA were more than twofold greater than the sum of the individual responses to peripheral and central chemoreflex stimulation alone (Jouett et al., 2015; Somers et al., 1989). Similarly, we observed that the MSNA and $\dot{V}_E$ responses to hypercapnic hypoxia were significantly greater than the physiological sum of the responses to isocapnic hypoxia and hyperoxic hypercapnia. This supports the concept that there is a hyper-additive relationship between the central and peripheral chemoreflexes in humans (Jouett et al., 2015; Somers et al., 1989). Our findings extend these previous studies by showing that the interaction between the peripheral and central chemoreflexes is not different in young women and men.

In contrast to MSNA, we observed that $\dot{V}_E$ responses to both hypercapnic hyperoxia and hypercapnic hypoxia were lower in women than men. The weak association between the ventilatory and sympathetic responses to chemoreflex activation has garnered recent attention (Keir et al., 2020; Prasad et al., 2020). This is partly because heightened chemoreflex tonicity and sensitivity have been identified in several patient populations (e.g. hypertension (Narkiewicz et al., 2016), heart failure (Narkiewicz, Pesek

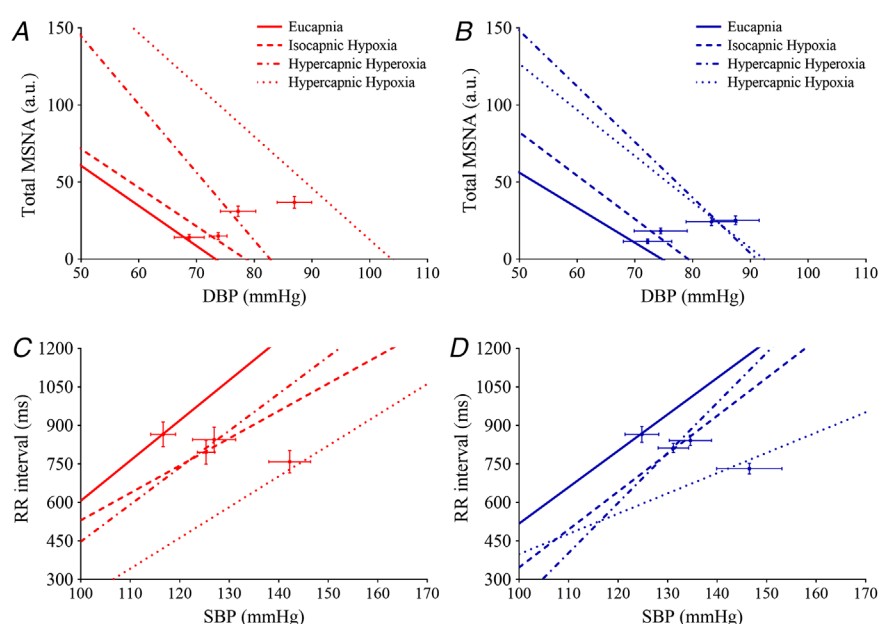

**Figure 4. Arterial baroreflex control of muscle sympathetic nerve activity (*A* and *B*) and R-R interval (*C* and *D*) during eucapnia, isocapnic hypoxia, hypercapnic hyperoxia and hypercapnic hypoxia in women (red triangles) and men (blue circles)**
Regression lines are displayed between DBP and total MSNA (*A* and *B*) and SBP and R-R interval (*C* and *D*) for eucapnia, isocapnic hypoxia, hypercapnic hyperoxia and hypercapnic hypoxia trials. Arterial baroreflex control of muscle sympathetic nerve activity where *n* = 9 women and *n* = 9 men (due to slopes $R^2 < 0.45$) and R-R interval where *n* = 7 women and *n* = 8 men (due to an absence of sequences). The comparison between sex and breathing trial was examined using a linear regression. [Colour figure can be viewed at wileyonlinelibrary.com]

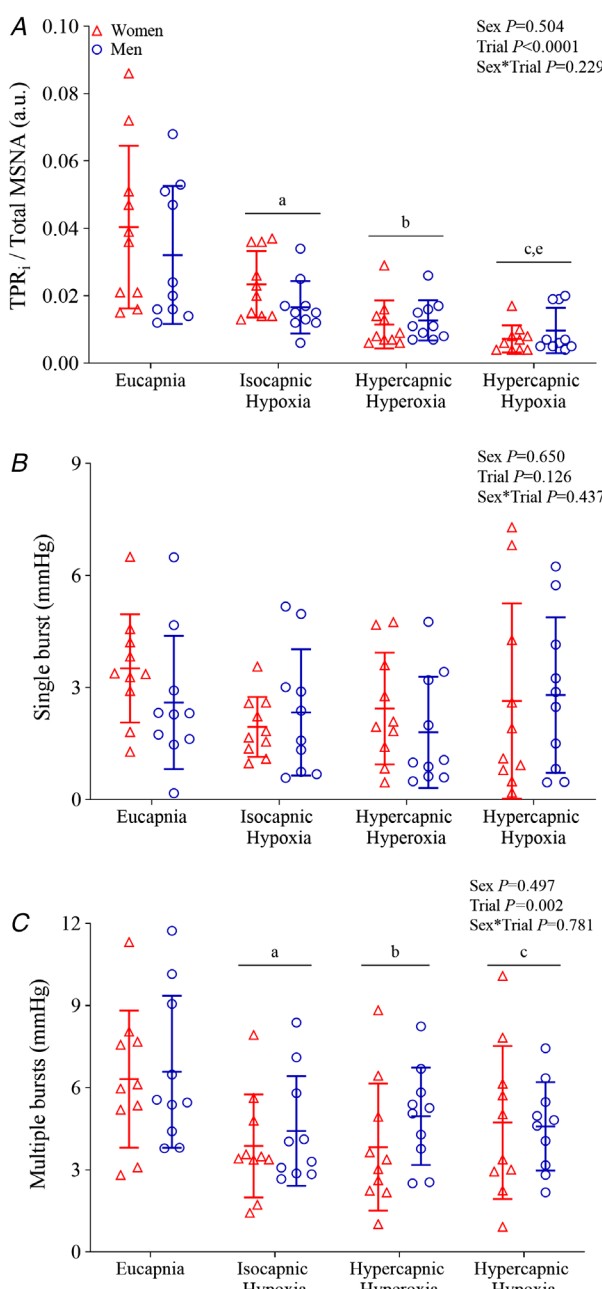

*post hoc* analysis (*t* tests with Bonferroni correction) are shown as [a]$P < 0.05$ eucapnia *vs.* isocapnic hypoxia, [b]$P < 0.05$ eucapnia *vs.* hypercapnic hyperoxia, [c]$P < 0.05$ eucapnia *vs.* hypercapnic hypoxia, [e]$P < 0.05$ isocapnic hypoxia *vs.* hypercapnic hypoxia. [Colour figure can be viewed at wileyonlinelibrary.com]

et al., 1999; Narkiewicz, van de Borne et al., 1999) and sleep apnoea (Li et al., 2016)), but in order to successfully identify patients in whom therapeutic targeting of the chemoreflexes might be most fruitful, characterization of the appropriate end-organ response is necessary (e.g. $\dot{V}_E$ *vs.* MSNA). Our results highlight that sex differences are an important consideration in this regard. There are several potential explanations for the lower $\dot{V}_E$ but greater MSNA responses we observe in women. It is possible that the augmented $\dot{V}_E$, and more specifically the $V_T$, responses in men causes sympatho-inhibition due to an augmented activation of the pulmonary stretch receptors (Seals et al., 1990). However, while changes in $V_T$ and breathing pattern reportedly alter the distribution of MSNA bursts over the respiratory cycle, MSNA burst frequency *per se* is not different (Fatouleh & Macefield, 2011; Limberg et al., 2013; Seals et al., 1990). Sex differences in the perception of the gas challenges (e.g. psychological stress) may have contributed to the differential responses (Anderson et al., 1987), but as women and men did not differ in their rating of perception of breathlessness this appears unlikely. Whether sex differences in cerebrovascular reactivity to the hypoxic and hypercapnic trials contributed to the ventilatory and sympathetic responses requires further investigation (Barnes & Charkoudian, 2021; Xie et al., 2006). Endogenous ovarian hormones have the potential to influence peripheral and central chemoreflex control of MSNA (Usselman et al., 2015), breathing (Behan & Wenninger, 2008) and cerebral blood flow (Barnes & Charkoudian, 2021). Previous studies examining the cerebral blood flow responses to hypercapnia are mixed, with greater (Kastrup et al., 1997), no different (Madureira et al., 2017) and attenuated (Kassner et al., 2010; Miller, Howery et al., 2019) responses reported in women. Unfortunately, cerebral blood flow was not measured in the current studies, but how any sex difference in cerebral blood flow might account for the differential MSNA (higher) and $\dot{V}_E$ (lower) responses to hypercapnic hyperoxia in women would be difficult to explain. Determining how menstrual cycle phase affects the integrated cardio-respiratory and neurovascular response to central and peripheral chemoreflex activation would help establish the mechanistic role of ovarian hormones in our observations.

## Arterial baroreflex function

The arterial baroreflex can restrain the MSNA response to chemoreflex activation (Heusser et al., 2020) and

**Figure 5. Sympathetic neurovascular transduction during eucapnia, isocapnic hypoxia, hypercapnic hyperoxia and hypercapnic hypoxia in women (red triangles, *n* = 10) and men (blue circles, *n* = 10)**

*A*, sympathetic neurovascular transduction expressed as the quotient of $TPR_i$ and total MSNA. *B*, the peak DBP responses following a single burst (occurring in isolation). *C*, the peak DBP responses following multiple bursts (adjacent to at least one other burst) during eucapnia, isocapnic hypoxia, hypercapnic hyperoxia and hypercapnic hypoxia. The main effects of sex, breathing trial and their interaction were examined using mixed model ANOVA with repeated measures. Where a significant interaction is observed, differences identified during *post hoc* analysis (*t* tests with Bonferroni correction) are identified as †$P < 0.05$ *vs.* eucapnia, ‡$P < 0.05$ *vs.* isocapnic hypoxia. Where a significant main effect of trial, but no interaction, is observed, differences observed during

**Table 3. Peak change in diastolic blood pressure following a cardiac cycle both with and without spontaneous muscle sympathetic nerve activity (MSNA) bursts, during eucapnia, isocapnic hypoxia, hypercapnic hyperoxia and hypercapnic hypoxia in women and men**

| | | Eucapnia | Isocapnic hypoxia | Hypercapnic hyperoxia | Hypercapnic hypoxia | ANOVA *P* values | | |
| | | | | | | Sex | Trial | Interaction |
|---|---|---|---|---|---|---|---|---|
| Cardiac cycle with MSNA bursts (mmHg) | Women | 5.4 ± 1.1 | 6.0 ± 1.4 | 4.4 ± 1.3 | 6.2 ± 1.7 | 0.819 | 0.021 | 0.917 |
| | Men | 5.7 ± 1.3 | 6.3 ± 1.9 | 5.5 ± 1.7 | 6.1 ± 2.3 | | a | |
| Cardiac cycle without MSNA bursts (mmHg) | Women | 5.3 ± 0.9 | 5.6 ± 1.6 | 4.7 ± 1.3 | 5.5 ± 2.2 | 0.682 | 0.161 | 0.399 |
| | Men | 5.0 ± 2.4 | 6.2 ± 1.5 | 3.7 ± 1.8 | 5.9 ± 1.8 | | | |

All variables are $n = 10$ women and $n = 10$ men. The main effects of sex, breathing trial and their interaction were examined using mixed model ANOVA with repeated measures. Where a significant main effect of trial, but no interaction, is observed, differences identified during *post hoc* analysis (*t* tests with Bonferroni correction) are shown as [a]$P < 0.05$ eucapnia *vs.* hypercapnic hyperoxia.

other physiological stressors (e.g. exercise; Scherrer et al., 1990) and as such, sex differences in arterial baroreflex control of MSNA provide another possible mechanism for the differential sympathetic responses to hypercapnic hyperoxia and hypercapnic hypoxia in men *vs.* women. In humans, both hypoxia and hypercapnia lead to a resetting of arterial baroreflex control of MSNA and HR to higher pressures with no changes in sensitivity (i.e. gain) (Halliwill & Minson, 2002; Steinback et al., 2009). Similarly, we observed that ABR-MSNA gain was unchanged during isocapnic hypoxia and hypercapnic hypoxia, although ABR-MSNA gain (Fig. 3) was increased during hypercapnic hyperoxia. Compared with eucapnia, cardiovagal indices of HRV were reduced during isocapnic hypoxia and hypercapnic hypoxia, possibly on account of both the direct effects of hypoxia and the increased $\dot{V}_E$ (Siebenmann et al., 2019). Importantly, no sex differences were identified in the arterial baroreflex control of MSNA, cBRS and HRV.

## Sympathetic neurovascular transduction

Despite apparent sex differences in the MSNA responses to hypercapnic hyperoxia and hypercapnic hypoxia, we observed no differences in neurovascular transduction between women and men in any breathing trial. Young women have previously been reported to exhibit smaller BP responses to spontaneously occurring sympathetic bursts than young men, indicative of blunted sympathetic neurovascular transduction (Robinson et al., 2019). Differences in the methodologies applied may explain these differing observations.

Sympathetic neurovascular transduction was assessed from the quotient of TPR$_i$ and total MSNA following the methods of Usselman et al. (2015) and observed to be decreased during isocapnic hypoxia, hypercapnic hyperoxia and hypercapnic hypoxia, compared with eucapnia. Neurovascular transduction was also assessed using a burst-triggered signal-averaging approach tracking

changes in BP following a spontaneous MSNA burst (Wallin & Nerhed, 1982). We observed that the magnitude of the peak DBP response following a cluster of MSNA bursts (i.e. more than one burst) was diminished during isocapnic hypoxia, hypercapnic hyperoxia and hypercapnic hypoxia, compared with eucapnia. However, when the peak DBP responses to single bursts were considered, no differences in neurovascular transduction breathing trials were observed, suggesting that a robust sympathetic stimulus was required for this to be manifested.

The effect of hypoxia on neurovascular transduction has been variously reported as being diminished (Heistad & Wheeler, 1970), preserved (Dinenno et al., 2003; Hansen et al., 2000; Marriott & Marshall, 1990; Rowell & Seals, 1990; Steele et al., 2021) and enhanced (Tan et al., 2013), likely on account of the varying experimental approaches employed and severity of hypoxia. Heistad & Wheeler (1970) reported that hypoxia significantly blunts the forearm vasoconstrictor responses to sympatho-excitation evoked by lower body negative pressure, a finding that contrasted with that of Rowell & Seals (1990) who employed a similar experimental approach. More recently, Dinenno et al. (2003) demonstrated that the local vasoconstrictor response to endogenous noradrenaline release evoked by tyramine was unchanged by hypoxia in men. We are not able to determine whether the apparent attenuation of sympathetic neurovascular transduction we observed during hypoxic conditions are due to diminished noradrenaline release (Leuenberger et al., 1991), desensitization of the vasculature to sympathetic stimulation (Vanhoutte et al., 1981), or upregulation of vasodilatory pathways (Blauw et al., 1995; Blitzer et al., 1996). Limited studies have examined how hypercapnia affects neurovascular transduction in humans. Using an animal model, McGillivray-Anderson & Faber (1990) showed that $CO_2$-induced acidosis inhibits skeletal muscle arteriolar vasoconstrictor responses to administration of a selective $\alpha_2$-agonist. This is consistent

**Table 4. Respiratory, cardiovascular and sympathetic variables during eucapnia, isocapnic hyperoxia (peripheral chemoreflex inhibition) and hypocapnic hyperoxia (central and peripheral chemoreflex inhibition) in women and men**

| | | Eucapnia | Isocapnic hyperoxia | Hypocapnic hyperoxia | ANOVA $P$ values Sex | Trial | Interaction |
|---|---|---|---|---|---|---|---|
| ***Respiration*** | | | | | | | |
| SpO$_2$ (%) | Women | 97 ± 1 | 99 ± 1 | 99 ± 1 | 0.960 | <0.0001 | 0.416 |
| | Men | 97 ± 1 | 99 ± 0 | 99 ± 0 | | [a,b] | |
| P$_{ET}$O$_2$ (mmHg) | Women | 97 ± 5 | 300 ± 28 | 320 ± 41 | 0.572 | <0.0001 | 0.464 |
| | Men | 96 ± 9 | 316 ± 31 | 320 ± 29 | | [a,b] | |
| P$_{ET}$CO$_2$ (mmHg) | Women | 41 ± 2 | 41 ± 4 | 36 ± 5 | 0.131 | <0.0001 | 0.560 |
| | Men | 42 ± 1 | 42 ± 3 | 37 ± 4 | | [b,c] | |
| $\dot{V}_E$ (l·min$^{-1}$) | Women | 12.1 ± 1.9 | 14.2 ± 4.2 | 18.1 ± 4.0 | 0.515 | <0.0001 | 0.749 |
| | Men | 13.9 ± 4.0 | 14.4 ± 3.9 | 18.9 ± 5.7 | | [b,c] | |
| V$_T$ (l) | Women | 0.93 ± 0.27 | 0.95 ± 0.37 | 0.76 ± 0.20 | 0.803 | 0.209 | 0.229 |
| | Men | 0.85 ± 0.22 | 0.95 ± 0.36 | 0.91 ± 0.26 | | | |
| R$f$ (breaths·min$^{-1}$) | Women | 13 ± 5 | 15 ± 4 | 23 ± 7 | 0.671 | <0.0001 | 0.114 |
| | Men | 15 ± 4 | 15 ± 5 | 19 ± 5 | | [b,c] | |
| Perception of breathlessness (a.u.) | Women | 0 ± 0 | 1 ± 2 | 1 ± 1 | 0.512 | 0.052 | 0.268 |
| | Men | 0 ± 0 | 0 ± 0 | 1 ± 0 | | | |
| ***Cardiovascular*** | | | | | | | |
| HR (beats·min$^{-1}$) | Women | 70 ± 10 | 65 ± 8 | 69 ± 8 | 0.666 | 0.002 | 0.857 |
| | Men | 69 ± 7 | 64 ± 5 | 67 ± 6 | | [a,c] | |
| SBP (mmHg) | Women | 113 ± 10 | 114 ± 11 | 121 ± 7 | 0.005 | 0.068 | 0.203 |
| | Men | 126 ± 10 | 131 ± 9 | 130 ± 10 | | | |
| DBP (mmHg) | Women | 71 ± 7 | 72 ± 9 | 76 ± 11 | 0.858 | 0.002 | 0.170 |
| | Men | 68 ± 12 | 75 ± 12 | 78 ± 9 | | [b] | |
| MAP (mmHg) | Women | 85 ± 7 | 87 ± 8 | 92 ± 12 | 0.149 | <0.0001 | 0.162 |
| | Men | 88 ± 12 | 97 ± 13 | 98 ± 11 | | [a,b] | |
| CO (l·min$^{-1}$) | Women | 5.6 ± 1.7 | 5.2 ± 1.3 | 5.5 ± 1.7 | 0.550 | 0.143 | 0.792 |
| | Men | 5.9 ± 1.2 | 5.7 ± 1.3 | 5.9 ± 1.4 | | | |
| SV (ml) | Women | 84 ± 20 | 84 ± 21 | 83 ± 25 | 0.585 | 0.563 | 0.778 |
| | Men | 87 ± 18 | 90 ± 23 | 88 ± 21 | | | |
| TPR (mmHg·l$^{-1}$·min$^{-1}$) | Women | 16.5 ± 5.4 | 17.6 ± 4.6 | 18.3 ± 6.6 | 0.932 | <0.0001 | 0.309 |
| | Men | 15.6 ± 4.5 | 18.3 ± 6.5 | 17.9 ± 5.7 | | [a,b] | |
| ***Sympathetic*** | | | | | | | |
| MSNA BF (bursts·min$^{-1}$) | Women | 11 ± 3 | 11 ± 4 | 12 ± 4 | 0.693 | 0.722 | 0.489 |
| | Men | 11 ± 5 | 13 ± 5 | 12 ± 4 | | | |
| MSNA BI (bursts·100 heartbeats$^{-1}$) | Women | 16 ± 6 | 17 ± 7 | 17 ± 6 | 0.694 | 0.225 | 0.574 |
| | Men | 17 ± 7 | 20 ± 8 | 18 ± 7 | | | |
| MSNA amplitude (%) | Women | 100 ± 0 | 133 ± 21 | 123 ± 29 | 0.063 | 0.023 | 0.201 |
| | Men | 100 ± 0 | 106 ± 27 | 114 ± 37 | | [a] | |
| Total MSNA (%) | Women | 100 ± 0 | 100 ± 27 | 95 ± 34 | 0.893 | 0.644 | 0.944 |
| | Men | 100 ± 0 | 102 ± 53 | 90 ± 31 | | | |

All variables are $n = 10$ women and $n = 10$ men. The main effects of sex, breathing trial and their interaction were examined using mixed model ANOVA with repeated measures. P$_{ET}$O$_2$, end-tidal oxygen; P$_{ET}$CO$_2$, end-tidal carbon dioxide; SpO$_2$, oxygen saturation; $\dot{V}_E$, minute ventilation; V$_T$, tidal volume; R$f$, breathing frequency; a.u., arbitrary units; HR, heart rate; SBP, systolic blood pressure; DBP, diastolic blood pressure; MAP, mean arterial pressure; CO, cardiac output; SV, stroke volume; TPR, total peripheral resistance; MSNA, muscle sympathetic nerve activity; BF, burst frequency; BI, burst incidence. Where a significant main effect of trial, but no interaction, is observed, differences identified during *post hoc* analysis (*t* tests with Bonferroni correction) are shown as [a] $P < 0.05$ eucapnia *vs.* isocapnic hyperoxia, [b] $P < 0.05$ eucapnia *vs.* hypocapnic hyperoxia, [c] $P < 0.05$ isocapnic hyperoxia *vs.* hypocapnic hyperoxia.

**Table 5. Baroreflex sensitivity and heart rate variability during eucapnia, isocapnic hyperoxia and hypocapnic hyperoxia in women and men**

| | | Eucapnia | Isocapnic hyperoxia | Hypocapnic hyperoxia | ANOVA P values | | |
| --- | --- | --- | --- | --- | --- | --- | --- |
| | | | | | Sex | Trial | Interaction |
| ***ABR-MSNA*** | | | | | | | |
| Burst incidence (bursts [100 heartbeats]$^{-1}$·mmHg$^{-1}$) | Women | $-4.14 \pm 1.45$ | $-4.35 \pm 1.87$ | $-4.29 \pm 2.63$ | 0.669 | 0.751 | 0.937 |
| | Men | $-3.80 \pm 1.54$ | $-4.22 \pm 2.53$ | $-3.98 \pm 1.48$ | | | |
| Total MSNA (a.u.·beat$^{-1}$·mmHg$^{-1}$) | Women | $-2.50 \pm 0.81$ | $-2.92 \pm 1.45$ | $-2.67 \pm 1.47$ | 0.548 | 0.335 | 0.997 |
| | Men | $-2.21 \pm 1.00$ | $-2.68 \pm 1.58$ | $-2.47 \pm 0.98$ | | | |
| ***cBRS*** | | | | | | | |
| Gain (ms·mmHg$^{-1}$) | Women | $15.6 \pm 5.0$ | $18.1 \pm 5.5$ | $8.5 \pm 2.9$ | 0.414 | 0.011 [b] | 0.233 |
| | Men | $14.2 \pm 5.3$ | $20.3 \pm 10.6$ | $15.0 \pm 10.8$ | | | |
| Number of sequences (*n*) | Women | $33 \pm 20$ | $18 \pm 16$ | $3 \pm 3$ | 0.884 | <0.0001 [a] | 0.144 |
| | Men | $23 \pm 19$ | $16 \pm 15$ | $12 \pm 12$ | | | |
| BEI (%) | Women | $0.57 \pm 0.32$ | $0.35 \pm 0.27$ | $0.13 \pm 0.10$ | 0.784 | 0.002 [a] | 0.391 |
| | Men | $0.50 \pm 0.25$ | $0.35 \pm 0.28$ | $0.28 \pm 0.20$ | | | |
| ***HRV*** | | | | | | | |
| RMSSD (ms) | Women | $73.1 \pm 41.2$ | $82.7 \pm 49.9$ | $55.0 \pm 31.4$ | 0.990 | 0.004 [b] | 0.583 |
| | Men | $69.2 \pm 34.9$ | $79.5 \pm 47.5$ | $62.8 \pm 49.1$ | | | |
| SDNN (ms) | Women | $77.8 \pm 31.4$ | $74.2 \pm 30.7$ | $55.1 \pm 18.8$ | 0.685 | <0.0001 [a,b] | 0.384 |
| | Men | $76.2 \pm 31.3$ | $83.8 \pm 38.8$ | $63.1 \pm 31.4$ | | | |
| HF (ms$^2$) | Women | $3223 \pm 4205$ | $3651 \pm 4502$ | $1047 \pm 979$ | 0.950 | 0.011 [b] | 0.504 |
| | Men | $2560 \pm 2170$ | $3281 \pm 3539$ | $1837 \pm 3137$ | | | |
| LF (ms$^2$) | Women | $2159 \pm 1948$ | $1366 \pm 1585$ | $643 \pm 560$ | 0.496 | 0.004 [a,b] | 0.231 |
| | Men | $2057 \pm 1909$ | $2610 \pm 3139$ | $988 \pm 921$ | | | |
| Total power | Women | $4874 \pm 4908$ | $3320 \pm 4859$ | $1535 \pm 1720$ | 0.701 | <0.0001 [a,b] | 0.17 |
| | Men | $4674 \pm 3971$ | $4548 \pm 6035$ | $2099 \pm 4006$ | | | |
| HF (n.u.) | Women | $60 \pm 27$ | $68 \pm 23$ | $63 \pm 20$ | 0.438 | 0.794 | 0.601 |
| | Men | $58 \pm 22$ | $56 \pm 23$ | $56 \pm 25$ | | | |
| LF (n.u.) | Women | $40 \pm 27$ | $30 \pm 22$ | $36 \pm 20$ | 0.397 | 0.691 | 0.508 |
| | Men | $42 \pm 22$ | $44 \pm 23$ | $44 \pm 25$ | | | |
| LF/HF ratio | Women | $2.62 \pm 4.78$ | $4.90 \pm 10.80$ | $1.25 \pm 1.90$ | 0.453 | 0.424 | 0.538 |
| | Men | $1.67 \pm 2.35$ | $1.97 \pm 2.27$ | $1.69 \pm 1.93$ | | | |

All variables are *n* = 10 women and *n* = 10 men, aside from cBRS where *n* = 7 women and *n* = 8 men (due to an absence of sequences). The main effects of sex, breathing trial and their interaction were examined using mixed model ANOVA with repeated measures. ABR-MSNA, arterial baroreflex control of MSNA; MSNA, muscle sympathetic nerve activity; cBRS, spontaneous cardiac baroreflex sensitivity; BEI, baroreflex effectiveness index; HRV, heart rate variability; RMSSD, square root of the mean of the sum of successive differences in R–R interval; SDNN, standard deviation of all normal sinus R–R intervals; HF, high frequency; n.u., normalized units; LF, low frequency. Where a significant main effect of trial, but no interaction, is observed, differences identified during *post hoc* analysis (*t* tests with Bonferroni correction) are shown as [a]$P < 0.05$ eucapnia *vs.* hypocapnic hyperoxia, [b]$P < 0.05$ isocapnic hyperoxia *vs.* hypocapnic hyperoxia.

with our data, showing blunted vascular responsiveness during hypercapnic conditions.

## Methodological considerations

Several experimental considerations should be acknowledged. We recognize the ongoing debate regarding the optimal method of assessing chemoreflex sensitivity and the relative strengths and weaknesses of the different methods available (Keir et al., 2020). We opted to use discrete steady-state gas trials in accordance with earlier work designed to test similar hypotheses to those of the present study (Somers et al., 1989) and to permit quantification of ABR-MSNA and neurovascular transduction. Previous studies have quantified the chemoreflex control of MSNA using a transient volitional breath-hold combined with hypoxia-hypercapnia to obviate the potential sympatho-inhibitory effects of augmented pulmonary afferent activation (Usselman et al., 2015). As participants breathed freely during our trials (aside from the hypocapnic hyperoxia trial), the possibility of sex differences in the sympatho-inhibitory effects of lung inflation cannot be discounted (Seals et al., 1990).

The duration of a hypoxic and hypercapnic gas exposure (seconds, minutes, hours, days) can evoke a marked effect on the physiological response and therefore the possibility cannot be ruled out that a different length of gas exposure could have evoked different responses. The duration of exposure and magnitude of hypoxic and hypercapnic stimuli were chosen based on previous work by Somers et al. (1989), in order to minimize participant discomfort and ensure that high quality MSNA recordings were maintained. To minimize the potential cofounding influence of prior gas exposure on the control of breathing and MSNA, trials were performed in a random order, separated by a $>10$ min washout period to ensure restoration of baseline room air values. Accordingly, resting MSNA, $\dot{V}_E$ and BP were not different prior to the start of each gas exposure (i.e. room air values).

Several of the trials we employed involved hyperoxia as it has been historically used to reduce peripheral chemoreflex activation (Dejours, 1962). However, hyperoxia is also known to evoke mild sympatho-excitation, vasoconstriction and hyperventilation (Becker et al., 1996), possibly on account of a direct central effect (Dean et al., 2004). We attempted to limit this in the present study by using only 50% $O_2$ and keeping the hyperoxic exposure to 5 min. Jones et al. (1999) previously observed an $\sim$20% reduction in total MSNA during 10 min hyperoxia (50% $F_iO_2$) in young men, while total MSNA remained unchanged in women. In contrast, we observed no reduction in total MSNA in either men or women. These observations are perhaps related to differences in the duration of the gas exposure and analytical approach used (minute by minute averages *vs*. 4 min average). $\dot{V}_E$ was not increased during our isocapnic hyperoxia control trial; however, MAP and TPR were elevated, possibly because of the increase in MSNA burst amplitude or a direct effect on the vasculature. Cardiovagal indices of HRV (RMSSD, SDNN, HF power) were reduced during hypocapnic hypoxia. However, given that hypocapnia was induced by guiding the participants to hyperventilate, we suspect that the altered cardiovagal control is not a direct effect of the hyperoxia but rather a secondary effect of the volitional increases in respiration that were necessary to induce the hypocapnia (Siebenmann et al., 2019). In contrast to some previous studies (Burt et al., 1995; Ng et al., 1993; Wiinberg et al., 1995), BP and MSNA were not different between women and men during eucapnia. However, an advantage of this is that baseline differences in these variables can be excluded as a mechanism for the differential responses to hypercapnic normoxia and hypercapnic hypoxia.

The Modelflow method was used to indirectly determine SV and shows good agreement with gold-standard techniques (e.g. thermodilution) (Harms et al., 1999; Jansen et al., 1990; Jellema et al., 1999; Matsukawa et al., 2004; Wesseling et al., 1993) and

has been extensively used (Badrov et al., 2017; Dujic et al., 2008; Kim et al., 2011; Vianna et al., 2012). This was used to calculate $TPR_i$, and as per the methods of Usselman et al. (2015), subsequently used to calculated as $TPR_i$/MSNA as an index of sympathetic transduction. However, to the best of our knowledge, the verification of this as an index of transduction has not been performed. Although we note that the neurovascular transduction responses reported using this metric were qualitatively similar to those derived from the magnitude of the peak DBP response following a cluster of MSNA bursts.

There is an ongoing debate regarding whether and how to control for the menstrual cycle phase, with the relevance to the research question stressed (Stanhewicz & Wong, 2020; Wenner & Stachenfeld, 2020). Herein, women were only studied during the early follicular phase due to logistical limitations and because it is at this phase that women are reported to exhibit an augmented MSNA response to an apnoea performed following the breathing of a hypercapnic hypoxic gas mixture (but not during the mid-luteal phase) (Usselman et al., 2015). Further investigations are required to determine whether our results would have differed had women been tested when ovarian hormone concentrations were higher, and more broadly to determine the influence of natural fluctuations in ovarian hormone concentrations on the integrated cardiorespiratory and sympathetic neurocirculatory responses to hypoxia and hypercapnia.

## Perspective

Chronic sympatho-excitation is implicated in the development of hypertension and a range of other pathophysiological processes (Fisher & Paton, 2012). Repeated central and peripheral chemoreflex activation has been implicated in both resting sympatho-excitation (Hedner et al., 1988) and raised BP (Carlson et al., 1993; Hla et al., 1994) in sleep apnoea syndrome. Herein we observed an augmented increase in MSNA in women in the early follicular phase of their menstrual cycle during both hypercapnic hypoxia and hypercapnic normoxia, but not isocapnic hypoxia, which points towards an augmented central chemoreflex control of MSNA in young women. However, a blunted sympathetic vascular transduction to high MSNA, which was observed in both sexes, means that such sympathetic responses do not translate into an augmented pressor response. Given that in post-menopausal women there is an increased prevalence of sleep apnoea and hypertension compared with men of a similar age (Redline et al., 1994), further studies are required to characterize the cardiorespiratory and sympathetic responses to peripheral, central and combined peripheral and central chemoreflex activation in older women.

## Summary

The objective of this study was to elucidate the differences in the sympathetic neurocirculatory responses to chemoreflex activation between young women and men. In accordance with earlier work (Usselman et al., 2015), young women displayed augmented increases in MSNA during combined central and peripheral chemoreflex activation in the early follicular phase of their menstrual cycle. Our findings show that this is attributable to an augmented central chemoreflex response, rather than an augmented peripheral chemoreflex response or a more marked hyper-additive response between the central and peripheral chemoreflexes. Whether this remains the case following menopause, when ovarian hormone concentrations are attenuated, remains to be determined.

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

## Additional information

### Data availability statement

Study data are available from the corresponding author upon reasonable request.

### Competing interests

None declared.

### Author contributions

A.L.S. contributed to the design of the work, to the acquisition, analysis, and interpretation of data, and to the drafting of the manuscript. J.-L.F. contributed to the data acquisition and to the revision of the manuscript. L.C.V. contributed to the analysis and interpretation of data, and to the revision of the manuscript. M.D. contributed to the design of the work, to data acquisition, and to the revision of the manuscript. J.F.R.P. contributed to the conception and design of the work, data interpretation, and to the revision of the manuscript. J.P.F. contributed to the conception and design of the work, to the acquisition, analysis, and interpretation of data, and to the writing of the manuscript. All authors approved the final version of the manuscript. All authors agree to be accountable for all aspects of the work in ensuring that questions related to the accuracy or integrity of any part of the work are appropriately investigated and resolved. All persons designated as authors qualify for authorship, and all those who qualify for authorship are listed.

### Funding

Support provided by Auckland Medical Research Foundation (Ref# 1119008. JPF, JFRP, ALCS), Health Research Council of New Zealand (Ref# 19/687. JPF, JFRP) and the Sydney Taylor Trust (JFRP).

### Acknowledgements

The authors wish to thank the participants for their enthusiastic participation in this study. The kind support of Auckland District Health Board is gratefully acknowledged.

Open access publishing facilitated by The University of Auckland, as part of the Wiley – The University of

Auckland agreement via the Council of Australian University Librarians.

## Keywords

blood pressure, central chemoreflex, neurovascular transduction, peripheral chemoreflex, sympathetic nervous system

## Supporting information

Additional supporting information can be found online in the Supporting Information section at the end of the HTML view of the article. Supporting information files available:

**Statistical Summary Document**
**Peer Review History**

