## [Peer Review History · The Journal of Physiology]

Sex-differences in the sympathetic neurocirculatory responses to chemoreflex activation

Ana Luiza C Sayegh, Jui-Lin Fan, Lauro C. Vianna, Mathew Dawes, Julian F. R. Paton, and James P Fisher
DOI: 10.1113/JP282327

Corresponding author(s): James Fisher (jp.fisher@auckland.ac.nz)

Review Timeline:

Submission Date:	01-Sep-2021
Editorial Decision:	09-Dec-2021
Revision Received:	02-Feb-2022
Editorial Decision:	29-Mar-2022
Revision Received:	13-Apr-2022
Accepted:	25-Apr-2022

Senior Editor: Harold Schultz

Reviewing Editor: Emma Hart

Transaction Report:

Dear Dr Dawes,

Re: JP-RP-2021-282327 "Sex-differences in the sympathetic neurocirculatory responses to chemoreflex activation" by Ana Luiza C Sayegh, Jui-Lin Fan, Lauro C. Vianna, Mathew Dawes, Julian F. R. Paton, and James P Fisher

Thank you for submitting your manuscript to The Journal of Physiology. It has been assessed by a Reviewing Editor and by 2 expert referees and I am pleased to tell you that it is considered to be acceptable for publication following satisfactory revision.

The reports are copied at the end of this email. Please address all of the points and incorporate all requested revisions, or explain in your Response to Referees why a change has not been made.

NEW POLICY: In order to improve the transparency of its peer review process The Journal of Physiology publishes online as supporting information the peer review history of all articles accepted for publication. Readers will have access to decision letters, including all Editors' comments and referee reports, for each version of the manuscript and any author responses to peer review comments. Referees can decide whether or not they wish to be named on the peer review history document.

I hope you will find the comments helpful and have no difficulty returning revisions within 4 weeks.

If you need to check to make sure that your Methods section conforms to the principles of UK regulations, you may wish to refer to Grundy (2015):

Grundy (2015) J. Physiol. 2015 Jun 15;593(12):2547-9 <https://doi.org/10.1113/JP270818>

Your revised manuscript should be submitted online using the links in Author Tasks Link Not Available. This link is to the Corresponding Author's own account, if this will cause any problems when submitting the revised version please contact us.

The image files from the previous version are retained on the system. Please ensure you replace or remove any files that have been revised.

REVISION CHECKLIST:

- Summary data must be reported as mean {plus minus} SD or 95% confidence interval
- All table and figure legends with summary data must include the statistical test used in the table/figure and sample size
- Figures with summary data bars must include individual data points, or box whisker plots when $n > 30$.
- Article file, including any tables and figure legends, must be in an editable format (eg Word)
- Upload each figure as a separate high quality file
- Upload a full Response to Referees, including a response to any Senior and Reviewing Editor Comments;
- Upload a copy of the manuscript with the changes highlighted.

- A potential 'Cover Art' file for consideration as the Issue's cover image;
- Appropriate Supporting Information (Video, audio or data set https://jp.msubmit.net/cgi-bin/main.plex?form_type=display_requirements#supp).

To create your 'Response to Referees' copy all the reports, including any comments from the Senior and Reviewing Editors, into a Word, or similar, file and respond to each point in colour or CAPITALS and upload this when you submit your revision.

I look forward to receiving your revised submission.

If you have any queries please reply to this email and the Peer Review Coordinator will be pleased to advise.

If revision is not possible, or if you cannot respond to the requests for change, contact us by return email as soon as

possible, giving reasons for the difficulties. Withdrawal of the manuscript may be necessary in these circumstances, and instruction will be given on how to proceed. Please note that a paper must be withdrawn before it can be submitted to another journal. If any issues remain unresolved please contact the Publications Office at jphysiol@physoc.org

If you would like help with English language editing, or other article preparation support, Wiley Editing Services offers expert help with English Language Editing, as well as translation, manuscript formatting, and figure formatting at www.wileyauthors.com/eeo/preparation. You can also check out our resources for Preparing Your Article for general guidance about writing and preparing your manuscript at www.wileyauthors.com/eeo/prepresources.

Yours sincerely,

Harold D Schultz
Senior Editor
The Journal of Physiology
<https://jp.msubmit.net>
<http://jp.physoc.org>
The Physiological Society
Hodgkin Huxley House
30 Farringdon Lane
London, EC1R 3AW
UK
<http://www.physoc.org>
<http://journals.physoc.org>

EDITOR COMMENTS

Reviewing Editor:

The authors aimed to examine whether there are sex differences in ventilatory, cardiovascular and sympathetic responses to central, peripheral and combined chemoreceptor activation. This is important because previous publications in this area have overlooked women and/or potential effects of sex hormones/genes on chemoreceptor control of the cardio-pulmonary systems. Showing that there is either a similar response in young women and men or a different response is important for future trials/studies going forward. The authors suggest that results indicate that there is a larger increase in MSNA (sympathetic nerve activity) in response to central chemoreceptor activation versus men, and that this was combined with a smaller increase in minute ventilation. This drove a similar response during hypercapnia hypoxia (combined activation; which is more physiologically relevant - as this would happen during apnoeas). However, there are some issues which need addressing - which are highlighted by both authors. My specific comments are below which overlap with some of the reviewers comments.

1. The paper is very long. The authors should define what their primary outcomes are and what their primary hypotheses are... what are they trying to find out here which is of primary importance? The important results are lost in the amount of data that is included. Perhaps some of the data could go into a supplement.
2. Physiological summation: to make it easier for the reader, can the authors clearly say what this is and why it is being done?
2. Is the duration of gas exposure adequate? Jones et al. 1999 indicate that longer exposures might be necessary to capture peak changes in MSNA. On the flip side - physiological relevance - apnoeas are short! Should exposures mimicking apnoea events be used instead?!
3. Are all changes compared to the preceding 5 mins before specific gas exposure or to the eucapnic exposure. Repeated exposure to different gas mixes could change the control of ventilation/MSNA etc - one could argue that the period

immediately before the specific gas exposure should be used.

4. TPR assessed from finger-photoplethysmography. What is the evidence that this is a valid measure of TPR? Does this measure even track the change in SV properly? Can the authors comment on this?

5. Again - can the authors in the methods provide support for their main outcome of sympathetic-vascular transduction - which used the TPR calculated from the finger pressure waveform.

6. Exact p-values need to be stated throughout.

7. I'm not 100% convinced that the augmented increase in MSNA in women during central stimulation is not driven by the smaller change in ventilation. I guess the interesting question here is why is there a smaller change in ventilation?

8. Since there is a lot of data here - are the authors going to make the data available for sharing to help maximise research into this area?

Senior Editor:

Comments for Authors to ensure the paper complies with the Statistics Policy:

Please state exact p values throughout including tables and figures (avoid using stat symbols in figures). State p values even when $P > 0.05$. Figure and table legends must state the statistical test used with the sample size and sample defined.

Comments to the Author:

Several composition, methodological and statistical issues were raised that need to be addressed. The Journal apologizes for the delay in this manuscript. Unfortunately, issues arose with a referee that were beyond their control and caused a delay. It was decided to allow more time rather than to recruit another reviewer (which would have caused a delay regardless).

REFeree COMMENTS

Referee #1:

This study assessed sex-differences in the MSNA and ventilatory response to activation of the central, peripheral, and combined central/peripheral chemoreceptors using a cohort of young, healthy men (n=10) and women (n=10). As the authors acknowledge, a comprehensive evaluation of sex differences in the effects of hypercapnia, hypoxia, and hypercapnic hypoxia on MSNA are limited. The inclusion of baroreflex sensitivity, sympathetic vascular transduction, and heart rate variability in addition to additional control conditions provides a comprehensive view of the integrative physiology. The approach is thorough and results add to a growing body of literature.

Major comments:

Based on results from Jones and colleagues (1999), why were protocols limited to 5-min? Jones et al showed women increase MSNA to hypoxia earlier (within 3 min) whereas men did not reach MSNA peak until 10 min of hypoxia. How might the shorter (5-min) protocol have affected observed sex differences and what might this mean for data interpretation?

It would be helpful for the authors to clarify why the early follicular phase was chosen. When considering mechanisms behind present findings, one might anticipate results would be enhanced if women were studied when hormone levels were higher. The authors should acknowledge within the discussion the findings from Usselman et al (2013 & 2015), which show sex differences are lost when women are studied with hormone levels at their peak. This is important when presuming sex differences mechanism would be lost post-menopause, despite no clear link between high hormones and enhanced chemoreflex-mediated increases in MSNA. This distinction should be outlined more clearly within the discussion.

The authors report a 2-way repeated measures ANOVA was used. Given the between (sex) by within (time) groups design, the appropriate statistical analysis is a mixed model ANOVA with repeated measures.

Please provide p-values for the statement at the end of "central and peripheral chemoreflexes: sympathetic responses" (Results page 15) for the physiological sum of changes in MSNA burst incidence vs responses to hypercapnic hypoxia. These data do not appear to be in the tables/figures and increases described within the text/Table 1 look similar between the physiological sum (Women: 15-19; Men: 14-15) and the response to hypercapnic hypoxia (Women: 20, Men: 12). Without this p-value, the authors main conclusions related to the facilitatory relationship between central/peripheral chemoreceptors on MSNA is based on Figure 2D where the effect of trial is $p=0.05$ (authors state $P<0.05$ is required for statistical significance).

Minor comments:

Although Jones and colleagues (1999) reported similar peak (%) increases in MSNA, based on lower MSNA in the women studied, the absolute increase in MSNA (which is how data from the current cohort are presented) was lower in the women. Although there were no observable group differences in resting MSNA, when authors analyze their data similar as a % change, are conclusions maintained?

Results from the hyperoxia trial disagree with data from Jones et al (1999) which found MSNA was reduced with hyperoxia in men only. Can the authors comment on differences in findings? In general, there is very limited discussion of results from the "control" protocols (Tables 4-5).

Although the authors acknowledge in the discussion (page 19) that the MSNA response to hypercapnic hypoxia was greater than the physiological sum of the response, there is no confirmation they also observed a greater ventilatory response. As a main finding of the study, this should be acknowledged somewhere.

Page 21 discussion states ABR MSNA gain (fig 3) was INCREASED during hypercapnic hyperoxia. Page 15 results states ABR-MSNA was REDUCED during hypercapnic hyperoxia vs eucapnia. There are no symbols within Table 2 or Figure 3 that show where the effect of trial is observed (no symbols on hypercapnic hypoxia). Please clarify.

Figures 3A and B seem to be missing MSNA error bars (y-axis)

The authors mention differences between studies may be due to the severity of hypoxia used (Page 23). Can the authors clarify how the level of hypoxia was chosen for the current investigation?

The meaning of "an more marked synergistic interaction" within the concluding paragraph (Page 26) is not inherently clear.

Can the authors comment on the race/ethnicity of the individuals studied?

The authors present a robust study in a clearly written manuscript that investigates sex differences in central, peripheral, and combined central and peripheral chemoreflex activation of sympathetic activity. It expands previous work by examining sex differences in neurovascular responses to central, peripheral, and combined central and peripheral chemoreflex activation in the same cohort. Strengths include combining central, peripheral, and combined chemoreflex activation in the same cohort, as well as thorough and transparent analyses/data presentation that include other measures of neurovascular control (baroreflex sensitivity, heart rate variability, neurovascular transduction). Limitations were minor and include studying women only in the early follicular phase of the menstrual cycle when sex hormones are at their lowest (may limit their ability to detect differences between sexes). Minor concerns are outlined below.

Methods:

-Were the participants on any medications other than the n=2 on oral contraceptives?

-Please provide justification for use of the early follicular phase- Historically this has been how menstrual cycle is controlled for in the literature, but moving forward (as discussed in a recent point-counterpoint in JAP) the research question should be considered when deciding if and how to control for menstrual cycle (PMIDs: 33197376, 32702274, 33197373, etc.). Are you limiting your ability to detect sex differences by testing female participants when sex steroids are at their lowest?

-Was 4 or 5min of data used for analysis in the baseline (eucapnia) trial? It appears that 5 min of baseline data is being compared to 4min of data during the other trials.

Results:

-Was there a complete dataset for each participant or did n vary by trial?

-p.15 top paragraph- does not mention the significant main effect of sex on SBP that is shown in Table 1

-Table 1- I am confused by the 100{plus minus}0 for burst amplitude for both men and women. If the largest burst under resting conditions is set as 100%, how is the average burst amplitude over the 4 or 5 min analyzed in the trial 100% as well?

-p.15, Central and peripheral chemoreflexes: sympathetic responses paragraph- Looking at the data, I agree that there seems to be a facilitatory interaction of central and peripheral chemoreflex responses in the women, but in the men the change in burst incidence was 4bursts/100hb for peripheral and 11 bursts/100hb for central (Δ 15bursts/100hb when added together), and change in burst incidence was 12bursts/100hb with the combined central and peripheral chemoreflex activation trial... not indicative of a facilitatory interaction in men.

-Figure 2. Are there statistical outliers driving or blunting any of the sex differences in total MSNA?

-Figure 3 legend- the other legends describe the individual data markers as women (red triangles) and men (blue circles). Please include the shapes in this figure legend as well, it reads women (red) and men (blue) which will be problematic if the reader prints in black and white.

-Figure legends for figures 4 and 5 are reversed (figure legend 4 describes fig 5 and vice versa).

-All Figure legends- consider including n of each sex per trial and information about statistical tests used to increase transparency.

Discussion

-p.18 first paragraph/p.19 2nd paragraph- I am not convinced that the MSNA response in men to hypercapnic hypoxia was greater than the physiological sum of the responses to isocapnic hypoxia and hyperoxic hypercapnia in this study.

-p. 24- Original citations should be used to reference to the previous work demonstrating sex differences in BP and MSNA.

Perspectives

-consider highlighting that the augmented increase in MSNA in women was seen in the early follicular phase of the menstrual cycle

-"in post-menopausal women there is an increased prevalence of sleep apnea and hypertension" - relative to what? Young women? Men of a similar age?

-This work is also interesting given the emerging data looking at sex differences in folks with sleep apnea

END OF COMMENTS

Confidential Review

01-Sep-2021

We are grateful to the Reviewing Editor, Senior Editor and both Referees for their careful consideration of our manuscript and the opportunity to submit a revised version. We believe that the insightful points raised have helped us to improve the clarity and impact of our work. As requested, we have provided a point-by-point response to the concerns raised (below) and both a clean version of our revised manuscript and a marked-up version showing the changes made in red underlined font (attached).

Reviewing Editor:

1. The paper is very long. The authors should define what their primary outcomes are and what their primary hypotheses are... what are they trying to find out here which is of primary importance? The important results are lost in the amount of data that is included. Perhaps some of the data could go into a supplement.

Response: We appreciate this feedback and accept the critique, although we respectfully believe that the comprehensive nature of our work is a strength. The stated primary hypothesis in the original manuscript was that “the central chemoreflex control of MSNA is enhanced in young women” and thus the main outcome variable is MSNA (muscle sympathetic nerve activity). As the Editor appreciates, in order to interpret this variable a number of other variables are also required to be presented (e.g., minute ventilation). As suggested, in the revised version of our manuscript we have emphasised the primary outcome and hypothesis. Moreover, secondary outcomes, namely arterial baroreflex sensitivity and sympathetic transduction, are clearly identified and the emphasis on these diminished. Specifically, we have deleted excess methodological information and the excluded secondary outcomes from key sections (Key Points, Abstract, first and last summary paragraphs of Discussion). We note that J Physiol, does not offer the option of including online supplementary materials (e.g., Figures, Tables, Methods) and we would prefer not to use a

remote depository to host figures etc. The changes described have reduced the length of our manuscript, however, the Referees have also requested several additions, which we have been happy to make but have lengthened several sections of our manuscript. Finally, we wish to include an additional figure to show original records from one individual so that the reader can appraise the quality of the signals obtained, and hope that this is satisfactory.

2. Physiological summation: to make it easier for the reader, can the authors clearly say what this is and why it is being done?

Response: Thank you for this suggestion. We regret that we were insufficiently clear on this important point. The “physiological sum” of the responses to isocapnic hypoxia (peripheral chemoreflex activation) and hypercapnic hyperoxia (central chemoreflex activation) was calculated for comparison with the response to hypercapnic hypoxia (combined peripheral and central chemoreflex sensitivity). This approach was used by Somers *et al.* (1989a) to explore the nature of the interaction between the central and peripheral chemoreflexes. If the response to hypercapnic hypoxia is not different to the sum of isocapnic hypoxia and hypercapnic hyperoxia, then it may be concluded that a simple algebraic summation of the central and peripheral chemoreflexes has occurred. However, if the response to hypercapnic hypoxia is greater than the sum of isocapnic hypoxia and hypercapnic hyperoxia, then it may be concluded that combined peripheral and central chemoreflex sensitivity evokes a hyper-additive response. Finally, if the sum is less than the summation of the component parts then an occlusive interaction may be concluded. This information has been added to the revised version of our manuscript (pages 11, lines 260-272).

3. Is the duration of gas exposure adequate? Jones *et al.* 1999 indicate that longer exposures might be necessary to capture peak changes in MSNA. On the flip side - physiologically relevance - apnoeas are short! Should exposures mimicking apnoea events be used instead?!

Response: The 5 min gas exposure duration that we used was chosen on the basis of the precedent set by Somers *et al.* (1989a) and with the logistical challenges in conducting the experiment in mind. It was considered that conducting each of the five gas exposures for 10 min (plus eucapnia/room air) would diminish our ability to maintain a high quality MSNA recording for the entire protocol versus making the experiment overly long for many participants, particularly once set-up time and adequate control and recovery times were factored in.

We agree that the duration of gas exposure (seconds, minutes, hours, days) can evoke a marked effect on the physiological response and cannot rule out the possibility that a different length of gas exposure could have evoked different responses. More specifically, we appreciate the apparent discrepancy between the data of Jones *et al.* (1999) and those of the current study, but as explained below (response to Referee #1 point 1) we do not believe that this explains why the findings of the two studies differ. Moreover, the more recent work of Miller *et al.* (2019) observed no differences in the MSNA response to hypoxia in young men and women, in agreement with the current study. The comparison of “*exposures mimicking apnoea events*” is an interesting suggestion and would make a good follow up study, however we respectfully feel that this is beyond the scope of the present investigation.

The rationale for our choice of gas exposure duration, and the associated strengths/weaknesses/alternative approaches, has been added to the revised version of our manuscript (page 23, line 536-541).

4. Are all changes compared to the preceding 5 mins before specific gas exposure or to the eucapnic exposure. Repeated exposure to different gas mixes could change the control of ventilation/MSNA etc - one could argue that the period immediately before the specific gas exposure should be used.

Response: All changes were compared to the eucapnic exposure. We accept that exposure to different gas mixtures could change the control of ventilation/MSNA etc. To limit this possibility, trials were performed in a random order and separated by >10 min to ensure restoration of baseline values. The success of the latter strategy is confirmed by the observation that no significant difference was found between the 1-min period prior to the start of each gas exposure trial (see Table 1 below). We have added a comment in the revised manuscript to address the limitation that repeated gas exposures may have affected the responses observed (page 23, lines 541-546).

Table 1: Selected physiological variables obtained during the eucapnia trial and in the 1 minute before each gas exposure trial (i.e., room air breathing).

	Eucapnia	Pre-Isocapnic Hypoxia	Pre-Hypercapnic Hyperoxia	Pre-Hypercapnic Hypoxia	Pre-Isocapnic Hyperoxia	Pre-Hypocapnic Hyperoxia	P
$P_{ET}O_2$ (mmHg)	100 ± 5	99 ± 6	98 ± 7	99 ± 5	100 ± 4	99 ± 7	0.82
$P_{ET}CO_2$ (mmHg)	42 ± 3	42 ± 3	42 ± 2	43 ± 3	43 ± 2	42 ± 2	0.89
VE (L.min ⁻¹)	13.5 ± 3.3	13.9 ± 3.1	13.1 ± 2.7	13.3 ± 2.6	13.6 ± 2.9	13.4 ± 3.0	0.93
MSNA BI (bursts·100 hb ⁻¹)	19 ± 8	18 ± 8	18 ± 7	19 ± 10	20 ± 9	19 ± 4	0.97
MAP (mmHg)	96 ± 10	97 ± 10	96 ± 11	96 ± 9	97 ± 11	97 ± 10	0.77

NB: No significant difference observed between the eucapnia trial and the 1-min prior to each gas trial (i.e., the last minute of the room air breathing recovery period of the previous trial) confirming that recovery periods were sufficient to permit restoration of resting values.

5. TPR assessed from finger-photoplethysmography. What is the evidence that this is a valid measure of TPR? Does this measure even track the change in SV properly? Can the authors comment on this?

Response: SV was determined indirectly by the ModelFlow method using finger-photoplethysmography. SV derived using this method shows good correspondence with

standard measurement techniques (e.g., thermodilution) (Jansen *et al.*, 1990; Wesseling *et al.*, 1993; Harms *et al.*, 1999; Jellema *et al.*, 1999; Matsukawa *et al.*, 2004) and has been extensively used in the research field (Dujic *et al.*, 2008; Kim *et al.*, 2011; Vianna *et al.*, 2012; Badrov *et al.*, 2017). This information, and the use of an indirect measure of SV (and thus TPR), has been added to the revised version of our manuscript (page 24, lines 568-574).

6. Again - can the authors in the methods provide support for their main outcome of sympathetic-vascular transduction - which used the TPR calculated from the finger pressure waveform.

Response: We have been more careful to denote sympathetic transduction as a secondary outcome in the revised manuscript. This was calculated as TPRi/MSNA following the methods of Usselman *et al.* (2015) to permit a direct comparison with their findings. To the best of our knowledge, the verification of this as an index of transduction (e.g., by studies employing more direct measures SV, such as thermodilution) have not been performed. Reassuringly, the sympathetic transduction responses reported using TPRi/MSNA were qualitatively similar that derived from the magnitude of the peak DBP response following a cluster of MSNA bursts. We have added this information to the revised manuscript (page 21, lines 495-497).

7. Exact p-values need to be stated throughout.

Response: Exact p-values are now stated throughout as requested.

8. I'm not 100% convinced that the augmented increase in MSNA in women during central stimulation is not driven by the smaller change in ventilation. I guess the interesting question here is why is there a smaller change in ventilation?

Response: We agree that this is an interesting question. We refer to evidence describing the potential effect of endogenous ovarian hormone concentrations on the control of breathing (Behan & Wenninger, 2008) and cerebral blood flow (Barnes & Charkoudian, 2021) in the revised manuscript (page 20, lines 456-457). Given the existing length of the manuscript, we have been concise and respectfully feel that further work is required before firm conclusions can be drawn.

9. Since there is a lot of data here - are the authors going to make the data available for sharing to help maximise research into this area?

Response: Data will be made available upon reasonable request to the corresponding author.

Senior Editor:

Comments for Authors to ensure the paper complies with the Statistics Policy:

Please state exact p values throughout including tables and figures (avoid using stat symbols in figures). State p values even when $P > 0.05$. Figure and table legends must state the statistical test used with the sample size and sample defined.

Response: Exact p-values are now stated throughout. Figure and table legends now state the statistical test used and sample size. We respectfully wish to keep the statistical symbols in the Figures, as without these we fear that the reader may be confused.

Referee #1:

Thank you for your excellent comments and suggestions. We have been able to respond to all of these positively.

Major comments:

1. Based on results from Jones and colleagues (1999), why were protocols limited to 5-min? Jones et al showed women increase MSNA to hypoxia earlier (within 3 min) whereas men did not reach MSNA peak until 10 min of hypoxia. How might the shorter (5-min) protocol have affected observed sex differences and what might this mean for data interpretation?

Response: As explained above to the Reviewing Editor, the 5 min gas trials were chosen following the method of Somers *et al.* (1989a) and with the logistical challenges in conducting the experiment in mind. When conceiving the study, we did not believe that conducting each of the 5 gas exposures for 10 min (plus eucapnia) would be practical, as we were concerned about participant comfort and the ability to maintain a high quality MSNA recording for the entire protocol (particularly once set-up, control and adequate recovery time were factored in). This rationale has been incorporated into the revised version of our manuscript (page 23, lines 536-546).

We note with interest the similar peak MSNA response, but shorter latency of response in women, with hypoxia in the study of Jones *et al.* (1999). Given that it was in the first 5 min of the Jones *et al.* (1999) study where sex-differences were observed, and that our gas exposures were 5 min in duration, it does not seem that the length of hypoxic exposure is the reason for the discrepant findings. Providing a definitive explanation for the different findings is difficult, however the lower MSNA in the women compared to men in Jones *et al.* (1999) (18 vs. 24 bursts/min) but not in the current study, the relatively small number of women Jones *et al.* (1999) studied (n=7), and differences in the phase of the menstrual cycle (early follicular phase vs. not controlled), may all be contributing factors.

Notably, in support of our observations, more recent work by Miller *et al.* (2019) reported no difference in MSNA during hypoxia in young men and women. Nevertheless, we accept that we cannot rule out the possibility that different responses would have been observed if a different length of gas exposure was administered. Accordingly, we have added

this as a study limitation and been clearer when describing observation of Jones *et al.* (1999) in the revised version of our manuscript (page 23, lines 536-539).

2. It would be helpful for the authors to clarify why the early follicular phase was chosen. When considering mechanisms behind present findings, one might anticipate results would be enhanced if women were studied when hormone levels were higher. The authors should acknowledge within the discussion the findings from Usselman et al (2013 & 2015), which show sex differences are lost when women are studied with hormone levels at their peak. This is important when presuming sex differences mechanism would be lost post-menopause, despite no clear link between high hormones and enhanced chemoreflex-mediated increases in MSNA. This distinction should be outlined more clearly within the discussion.

Response: Thank you for this important point. Due to logistical reasons, we were only able to test women at one timepoint and considered that the early follicular phase would represent a good starting point at which ovarian hormone concentrations would be most similar among the women studied. Moreover, as the Referee states, it was during the early follicular phase (but not midluteal phase) that Usselman et al., observed sex-differences in the MSNA response to an apnoea performed following the breathing of a hypercapnic hypoxic gas mixture. We agree that our results might have been different if women were studied when ovarian hormone concentrations were higher, and it would be interesting to investigate this possibility. The manuscript has been revised to acknowledge this issue and the important work highlighted (page 25, lines 578-586).

3. The authors report a 2-way repeated measures ANOVA was used. Given the between (sex) by within (time) groups design, the appropriate statistical analysis is a mixed model ANOVA with repeated measures.

Response: A mixed model ANOVA with repeated measures was used. Apologies for not being clearer. This has been corrected (page 12, line 278-279).

4. Please provide p-values for the statement at the end of “central and peripheral chemoreflexes: sympathetic responses” (Results page 15) for the physiological sum of changes in MSNA burst incidence vs responses to hypercapnic hypoxia. These data do not appear to be in the tables/figures and increases described within the text/Table 1 look similar between the physiological sum (Women: 15-19; Men: 14-15) and the response to hypercapnic hypoxia (Women: 20, Men: 12). Without this p-value, the authors’ main conclusions related to the facilitatory relationship between central/peripheral chemoreceptors on MSNA is based on Figure 2D where the effect of trial is $p=0.05$ (authors state $P<0.05$ is required for statistical significance).

Response: We apologise for the confusion resulting from our typographical error. Rather than referring to MSNA burst incidence we should have focused on total MSNA as per Figure 3 (Figure 2 in the original submission). A focus is placed on total MSNA as it has been identified as a more sensitive measure of the sympathetic responses to physiological stimuli (Rowell & Blackmon, 1987; Jones *et al.*, 1999). In the revised version of the manuscript, we have included the p-value for the physiological sum of the total MSNA responses to isocapnic hypoxia and hypercapnic hyperoxia compared to hypercapnic hypoxia in women and men (page 14, line 326).

Minor comments:

1. Although Jones and colleagues (1999) reported similar peak (%) increases in MSNA, based on lower MSNA in the women studied, the absolute increase in MSNA (which is how data from the current cohort are presented) was lower in the women. Although there were no

observable group differences in resting MSNA, when authors analyzed their data similar as a % change, are conclusions maintained?

Response: To the best of our knowledge, our data are already presented in a similar way as Jones *et al.* (1999) (i.e., as a percentage change in total MSNA), although we denote eucapnia as 100 (our Figure 3C; Figure 2C in the original submission) whereas Jones *et al.* (1999) denote this as 0 (their Figure 3). Please note that the data analysis section of the Jones *et al.* study is brief, so details are sparse.

2. Results from the hyperoxia trial disagree with data from Jones et al (1999) which found MSNA was reduced with hyperoxia in men only. Can the authors comment on differences in findings? In general, there is very limited discussion of results from the "control" protocols (Tables 4-5).

Response: This is a good point. Due to the length of the manuscript we tried to keep the focus on the main outcome variables. However, as requested we have added a brief comment about the differing responses to hyperoxia in the current work and those of Jones *et al.* (1999) (page 19, line 427-431).

3. Although the authors acknowledge in the discussion (page 19) that the MSNA response to hypercapnic hypoxia was greater than the physiological sum of the response, there is no confirmation they also observed a greater ventilatory response. As a main finding of the study, this should be acknowledged somewhere.

Response: Thank you for this suggestion. This information has been added (page 17, line 383 and page 18, line 420).

4. Page 21 discussion states ABR MSNA gain (fig 3) was *INCREASED* during hypercapnic hyperoxia. Page 15 results states ABR-MSNA was *REDUCED* during hypercapnic hyperoxia vs eucapnia. There are no symbols within Table 2 or Figure 3 that show where the effect of trial is observed (no symbols on hypercapnic hypoxia). Please clarify.

Response: Thank you for pointing out this typographical error. The discussion section has been corrected to state that ABR-MSNA gain was decreased during hypercapnic hyperoxia vs eucapnia (page 21, line 477). Regarding the symbols in Table 2 and Figure 4 (Figure 3 in the original submission), these were only included to denote significant *post hoc* differences where there is a significant interaction between sex and trial. *Post hoc* differences following a significant main effect of trial are described in the Results text. This is now clarified in the Figure and Table legends were appropriate.

5. Figures 3A and B seem to be missing MSNA error bars (y-axis)

Response: Thank you for pointing this out. The symbol size in Figure 4 (Figure 3 in the original submission) has been reduced so the error bars are now clear.

6. The authors mention differences between studies may be due to the severity of hypoxia used (Page 23). Can the authors clarify how the level of hypoxia was chosen for the current investigation?

Response: The severity of hypoxia used was chosen based on the work of (Somers *et al.*, 1989a). This degree of hypoxia is commonly used in both healthy participants and patients with chronic diseases (Somers *et al.*, 1988; Somers *et al.*, 1989b; Leuenberger *et al.*, 2005; Foster *et al.*, 2008; Miller *et al.*, 2019). As we included both isocapnic hypoxia and hypercapnic hypoxia trials and needed to use the same hypoxic stimulus in both, we were conscious that trials should be tolerated by participants. That is to say, when hypoxia is

combined with hypercapnia it is a stronger stimulus than when delivered alone, therefore using severe hypoxia per se would not be tolerable for all participants (i.e., during hypercapnic hypoxia). A comment on this issue has been added (page 23, line 539-541).

7. The meaning of "a more marked synergistic interaction" within the concluding paragraph (Page 26) is not inherently clear.

Response: This has been replaced with “a more marked hyper-additive response” (page 26, line 612).

8. Can the authors comment on the race/ethnicity of the individuals studied?

Response: Of the individuals studied, 4 identified as Middle Eastern/Latin American/African (2 women), 5 identified as Asian (2 women) and 11 identified as European (6 women). These race/ethnicity categories are those used by the New Zealand census. This information has now been added to the manuscript (Page 7, line 148-150). We are not sufficiently powered to perform a sub-analysis based on ethnicity.

Referee #2:

Methods:

1. -Were the participants on any medications other than the n=2 on oral contraceptives?

Response: Aside from oral contraceptives, participants were not taking any medications (page 7, line 154).

2. -Please provide justification for use of the early follicular phase- Historically this has been how the menstrual cycle is controlled for in the literature, but moving forward (as discussed in a recent point: counterpoint in JAP) the research question should be considered when

deciding if and how to control for the menstrual cycle (PMIDs:33197376, 32702274, 33197373, etc.). Are you limiting your ability to detect sex differences by testing female participants when sex steroids are at their lowest?

Response: This is an important point, and we agree with the Referee that the research question should inform the decision as to if/how menstrual cycle is controlled. As mentioned to Referee #1, due to logistical reasons we were only able to test women at one timepoint and it was during this early follicular phase (but not midluteal phase) that Usselman et al., observed sex-differences in the MSNA response to an apnoea performed following the breathing of a hypercapnic hypoxic gas mixture. While it is possible that we may be limiting our ability to detect sex differences by studying the early follicular phase, it is important to note that several fundamental differences between women and men were identified. Nevertheless, it is tempting to speculate that our results might have been different had women been studied when ovarian hormone concentrations were higher. A comment on this issue has been added to the revised manuscript (page 25, lines 578-586).

3. Was 4 or 5 min of data used for analysis in the baseline (eucapnia) trial? It appears that 5 min of baseline data is being compared to 4 min of data during the other trials.

Response: Cardiorespiratory and sympathetic variables were averaged over the last four minutes of each trial (including eucapnic) (page 9, line 203).

Results:

4. -Was there a complete dataset for each participant or did n vary by trial?

Response: There is a complete dataset for each participant for all the main outcome variables. The exception to this was 1 woman and 1 man for the ABR-MSNA analysis (slopes $R^2 < 0.45$) and 3 women and 2 men from the cBRS analysis (no sequences).

Participant numbers have been added to each Figure and Table legend in the revised manuscript.

5. *-p.15 top paragraph- does not mention the significant main effect of sex on SBP that is shown in Table 1*

Response: This paragraph now also describes the SBP responses (page 13, lines 306-308).

6. *-Table 1- I am confused by the 100 (plus minus) 0 for burst amplitude for both men and women. If the largest burst under resting conditions is set as 100%, how is the average burst amplitude over the 4 or 5 min analyzed in the trial 100% as well?*

Response: Thank you for the opportunity to clarify the analytical approach used. As is convention, we identified the largest burst occurring for a given participant during eucapnia. This burst was assigned a value of 100, and all other bursts were expressed relative to this. An average burst amplitude for a given individual for eucapnia (and subsequently all other gas exposures) can then be calculated. For the summary data presented in Table 1, the eucapnia values is assigned 100% and the change in amplitude for the other gas conditions expressed relative to this (i.e., in percent). We feel that the normalisation of burst amplitude in this way removes artificial differences that may occur between groups (i.e., associated with the proximity of the microelectrode to the sympathetic fascicle).

7. *-p.15, Central and peripheral chemoreflexes: sympathetic responses paragraph- Looking at the data, I agree that there seems to be a facilitatory interaction of central and peripheral chemoreflex responses in the women, but in the men the change in burst incidence was 4bursts/100hb for peripheral and 11 bursts/100hb for central (Δ 15bursts/100hb when added*

together), and change in burst incidence was 12bursts/100hb with the combined central and peripheral chemoreflex activation trial... not indicative of a facilitatory interaction in men.

Response: Thank you for raising this point. We apologise for the confusion resulting from our typographical error. The physiological sum of the changes in total MSNA was lower than hypercapnic hypoxia in women and men as per Figure 3 (Figure 2 in the original submission). A reliance is placed on total MSNA as it has been considered to be a more sensitivity measure of the sympathetic responses to physiological stimuli (Rowell & Blackmon, 1987; Jones *et al.*, 1999). We changed the data description from MSNA burst incidence to Total MSNA in the results section (page 14, line 325-330).

8. -Figure 2. Are there statistical outliers driving or blunting any of the sex differences in total MSNA?

Response: The data in Figure 3 (Figure 2 in the original submission) technically do not contain statistical outliers (i.e., values greater than two standard deviations from the mean). However, 1 woman and 1 man do deviate quite a lot from the mean for total MSNA, but the results are unchanged when these data are omitted.

9. -Figure 3 legend- the other legends describe the individual data markers as women (red triangles) and men (blue circles). Please include the shapes in this figure legend as well, it reads women (red) and men (blue) which will be problematic if the reader prints in black and white.

Response: Done.

10. -Figure legends for figures 4 and 5 are reversed (figure legend 4 describes fig 5 and vice versa).

Response: We thank the Referee for pointing this out. This has been corrected.

11. -All Figure legends- consider including n of each sex per trial and information about statistical tests used to increase transparency.

Response: The requested information has been added.

Discussion

12. -p.18 first paragraph/p.19 2nd paragraph- I am not convinced that the MSNA response in men to hypercapnic hypoxia was greater than the physiological sum of the responses to isocapnic hypoxia and hyperoxic hypercapnia in this study.

Response: As shown in Figure 3 (Figure 2 in the original submission), the total MSNA response to hypercapnic hypoxia was significantly greater than the physiological sum of the responses to isocapnic hypoxia and hypercapnic hyperoxia in men. A reliance is placed on total MSNA as it has been considered to be a more sensitivity measure of the sympathetic responses to physiological stimuli (Rowell & Blackmon, 1987; Jones *et al.*, 1999). As mentioned above, due to a typographical error the results section of the originally submitted manuscript included MSNA burst incidence. However, in the revised manuscript total MSNA is provided in the results section (page 14, line 325-330).

13. -p. 24- Original citations should be used to reference to the previous work demonstrating sex differences in BP and MSNA.

Response: Done.

Perspectives

14. -consider highlighting that the augmented increase in MSNA in women was seen in the early follicular phase of the menstrual cycle

Response: Thank you for this suggestion. We have included this information in the “Perspective” (page 25, line 594) and “Summary” (page 26, line 609) sections.

15. -"in post-menopausal women there is an increased prevalence of sleep apnea and hypertension" - relative to what? Young women? Men of a similar age?

Response: Women have an increased prevalence of sleep apnoea and hypertension relative to men of a similar age. This has been clarified in the revised manuscript (page 26, line 599).

16. -This work is also interesting given the emerging data looking at sex differences in folks with sleep apnea

Response: We agree with the Referee that this is interesting.

References

Badrov MB, Barak OF, Mijacika T, Shoemaker LN, Borrell LJ, Lojpur M, Drvis I, Dujic Z & Shoemaker JK. (2017). Ventilation inhibits sympathetic action potential recruitment even during severe chemoreflex stress. *J Neurophysiol* **118**, 2914-2924.

Barnes JN & Charkoudian N. (2021). Integrative cardiovascular control in women: Regulation of blood pressure, body temperature, and cerebrovascular responsiveness. *FASEB J* **35**, e21143.

Behan M & Wenninger JM. (2008). Sex steroidal hormones and respiratory control. *Respir Physiol Neurobiol* **164**, 213-221.

Dujic Z, Ivancev V, Heusser K, Dzamonja G, Palada I, Valic Z, Tank J, Obad A, Bakovic D, Diedrich A, Joyner MJ & Jordan J. (2008). Central chemoreflex sensitivity and sympathetic neural outflow in elite breath-hold divers. *J Appl Physiol (1985)* **104**, 205-211.

Foster GE, Hanly PJ, Ostrowski M & Poulin MJ. (2008). Ventilatory and blood pressure responses to isocapnic hypoxia in OSA patients. *Adv Exp Med Biol* **605**, 463-468.

- Harms MP, Wesseling KH, Pott F, Jenstrup M, Van Goudoever J, Secher NH & Van Lieshout JJ. (1999). Continuous stroke volume monitoring by modelling flow from non-invasive measurement of arterial pressure in humans under orthostatic stress. *Clin Sci (Lond)* **97**, 291-301.
- Jansen JR, Wesseling KH, Settels JJ & Schreuder JJ. (1990). Continuous cardiac output monitoring by pulse contour during cardiac surgery. *Eur Heart J* **11 Suppl I**, 26-32.
- Jellema WT, Wesseling KH, Groeneveld AB, Stoutenbeek CP, Thijs LG & van Lieshout JJ. (1999). Continuous cardiac output in septic shock by simulating a model of the aortic input impedance: a comparison with bolus injection thermodilution. *Anesthesiology* **90**, 1317-1328.
- Jones PP, Davy KP & Seals DR. (1999). Influence of gender on the sympathetic neural adjustments to alterations in systemic oxygen levels in humans. *Clin Physiol* **19**, 153-160.
- Kim A, Deo SH, Vianna LC, Balanos GM, Hartwich D, Fisher JP & Fadel PJ. (2011). Sex differences in carotid baroreflex control of arterial blood pressure in humans: relative contribution of cardiac output and total vascular conductance. *Am J Physiol Heart Circ Physiol* **301**, H2454-2465.
- Leuenberger UA, Brubaker D, Quraishi SA, Hogeman CS, Imadojemu VA & Gray KS. (2005). Effects of intermittent hypoxia on sympathetic activity and blood pressure in humans. *Auton Neurosci* **121**, 87-93.
- Matsukawa K, Kobayashi T, Nakamoto T, Murata J, Komine H & Noso M. (2004). Noninvasive evaluation of cardiac output during postural change and exercise in humans: comparison between the modelflow and pulse dye-densitometry. *Jpn J Physiol* **54**, 153-160.
- Miller AJ, Cui J, Luck JC, Sinoway LI & Muller MD. (2019). Age and sex differences in sympathetic and hemodynamic responses to hypoxia and cold pressor test. *Physiol Rep* **7**, e13988.
- Rowell LB & Blackmon JR. (1987). Human cardiovascular adjustments to acute hypoxaemia. *Clin Physiol* **7**, 349-376.
- Somers VK, Mark AL & Abboud FM. (1988). Potentiation of sympathetic nerve responses to hypoxia in borderline hypertensive subjects. *Hypertension* **11**, 608-612.
- Somers VK, Mark AL, Zavala DC & Abboud FM. (1989a). Contrasting effects of hypoxia and hypercapnia on ventilation and sympathetic activity in humans. *J Appl Physiol* (1985) **67**, 2101-2106.
- Somers VK, Mark AL, Zavala DC & Abboud FM. (1989b). Influence of ventilation and hypocapnia on sympathetic nerve responses to hypoxia in normal humans. *J Appl Physiol* (1985) **67**, 2095-2100.

- Usselman CW, Gimon TI, Nielson CA, Luchyshyn TA, Coverdale NS, Van Uum SH & Shoemaker JK. (2015). Menstrual cycle and sex effects on sympathetic responses to acute chemoreflex stress. *Am J Physiol Heart Circ Physiol* **308**, H664-671.
- Vianna LC, Hart EC, Fairfax ST, Charkoudian N, Joyner MJ & Fadel PJ. (2012). Influence of age and sex on the pressor response following a spontaneous burst of muscle sympathetic nerve activity. *Am J Physiol Heart Circ Physiol* **302**, H2419-2427.
- Wesseling KH, Jansen JR, Settels JJ & Schreuder JJ. (1993). Computation of aortic flow from pressure in humans using a nonlinear, three-element model. *J Appl Physiol* (1985) **74**, 2566-2573.

Dear Dr Fisher,

Re: JP-RP-2022-282327R1 "Sex-differences in the sympathetic neurocirculatory responses to chemoreflex activation" by Ana Luiza C Sayegh, Jui-Lin Fan, Lauro C. Vianna, Mathew Dawes, Julian F. R. Paton, and James P Fisher

Please accept our apologies for the delay in providing you with an editorial decision on your revised paper.

Thank you for submitting your manuscript to The Journal of Physiology. It has been assessed by a Reviewing Editor and by 2 expert Referees and I am pleased to tell you that it is considered to be acceptable for publication following satisfactory revision.

The reports are copied at the end of this email. Please address all of the points and incorporate all requested revisions, or explain in your Response to Referees why a change has not been made.

NEW POLICY: In order to improve the transparency of its peer review process The Journal of Physiology publishes online as supporting information the peer review history of all articles accepted for publication. Readers will have access to decision letters, including all Editors' comments and referee reports, for each version of the manuscript and any author responses to peer review comments. Referees can decide whether or not they wish to be named on the peer review history document.

Authors are asked to use The Journal's premium BioRender (<https://biorender.com/>) account to create/redraw their Abstract Figures. Information on how to access The Journal's premium BioRender account is here: <https://physoc.onlinelibrary.wiley.com/journal/14697793/biorender-access> and authors are expected to use this service. This will enable Authors to download high-resolution versions of their figures. The link provided should only be used for the purposes of this submission. Authors will be charged for figures created on this premium BioRender account if they are not related to this manuscript submission.

I hope you will find the comments helpful and have no difficulty returning your revisions within 4 weeks.

Your revised manuscript should be submitted online using the links in Author Tasks Link Not Available.

Any image files uploaded with the previous version are retained on the system. Please ensure you replace or remove all files that have been revised.

REVISION CHECKLIST:

- Article file, including any tables and figure legends, must be in an editable format (eg Word)
- Abstract figure file (see above)
- Statistical Summary Document
- Upload each figure as a separate high quality file
- Upload a full Response to Referees, including a response to any Senior and Reviewing Editor Comments;
- Upload a copy of the manuscript with the changes highlighted.

- A potential 'Cover Art' file for consideration as the Issue's cover image;
- Appropriate Supporting Information (Video, audio or data set https://jp.msubmit.net/cgi-bin/main.plex?form_type=display_requirements#supp).

To create your 'Response to Referees' copy all the reports, including any comments from the Senior and Reviewing Editors, into a Word, or similar, file and respond to each point in colour or CAPITALS and upload this when you submit your revision.

I look forward to receiving your revised submission.

If you have any queries please reply to this email and staff will be happy to assist.

Yours sincerely,

Harold D Schultz
Senior Editor
The Journal of Physiology
<https://jp.msubmit.net>
<http://jp.physoc.org>
The Physiological Society
Hodgkin Huxley House
30 Farringdon Lane
London, EC1R 3AW
UK
<http://www.physoc.org>
<http://journals.physoc.org>

REQUIRED ITEMS:

The Journal of Physiology funds authors of provisionally accepted papers to use the premium BioRender site to create high resolution schematic figures. Follow this link and enter your details and the manuscript number to create and download figures. Upload these as the figure files for your revised submission. If you choose not to take up this offer we require figures to be of similar quality and resolution. If you are opting out of this service to authors, state this in the Comments section on the Detailed Information page of the submission form.

EDITOR COMMENTS

Reviewing Editor:

Thank you for re-submitting this manuscript. The manuscript is much improved. However the readability is difficult in places and reviewer 1 has pointed out some places where this could be improved. There are also some inconsistencies within the manuscript which are pointed out by reviewer 1, such as inconsistencies in p-values. Please also check that exact p-values are reported: in the abstract there is a p-value reported as $P > 0.05$

Senior Editor:

There is still a problem with reporting p values. The exact p values need to be reported in the tables.

In the figures, please report the exact p values for trials. The only exception to this is if p is less than 0.0001, in which case '<' is permitted.

In the figure legends, please include: The exact p values for comparisons are shown in the statistical summary document.

If the statistical summary document has errors please describe what is incorrect. (Required):

The statistical summary document is not complete. All of the statistical comparisons shown in the figures need to be included in the document with the exact p values. This is particularly important since the authors chose to continue to use symbols in the figures.

In the results text, statistical comparisons not shown in tables or figures need to be included in the stat summary document.

REFeree COMMENTS

Referee #1:

The authors did a very nice job addressing prior comments. The comprehensiveness of the data collected will significantly advance knowledge within the field. In reviewing the revision, some minor issues should be addressed to ensure validity of conclusions and readability.

The authors outline a number of post-hoc analyses by trial within the results section, however these trial differences are not acknowledged within the tables. It would be helpful to indicate within the tables, using relevant symbols, results of the post-hoc trial comparisons.

The authors should consider whether the terms men/women or male/female are more appropriate when describing their study populations. PMID: 34797173

Figure 6 should include the statistical analysis used and the results, just as was done in the other figures.

The authors state the BRS gain is reduced (Line 340). However, it appears the negative slope becomes more negative (supporting greater change in MSNA per unit change in BP). This reviewer interprets the results to show increased gain. This should be corrected throughout.

In the methods Page 11, the authors describe BEI (baroreflex effectiveness index). This is not reported anywhere and should be removed.

Within the abstract, the authors use $p=0.05$ to describe sex differences in the MSNA responses. It is unclear what this is referring to (MSNA amp? total MSNA?). These p-values could not be found within the body of the manuscript. The p-values/data reported in the abstract should be consistent with what is reported within the manuscript (perhaps total MSNA, $p=0.02$ and $p<0.001$; Lines 329-330).

Line 307 states perception of breathlessness changed across trials, but was not different between men and women ($p=0.70$). The interaction value reported in Table 1 is $p=0.86$. What p-value are the authors referring to?

Figure 3D, which supports a trial effect (hyper-additive) is $p=0.05$. According to the results section, should this be $p=0.045$?

This author appreciates acknowledging differences between hyperoxia trials with Jones et al (Lines 433-437). This may be more appropriately placed with discussion of method limitations where hyperoxia is similarly discussed (Page 23, Lines 554-561).

The authors acknowledge women were studied only one (Lines 571-574, Page 24). This may be more appropriately added to the other section discussing menstrual cycle (Line 586, Page 25)

Table 1: Men MSNA amplitude during hypercapnic hyperoxia is missing a symbol showing increase from eucapnia.

Line 483 and Line 484-485, "ABR-MSNA gain is unchanged during hypercapnic hypoxia" is stated twice.

The authors may want to include new reference PMID: 34528146 when discussing hypoxia and transduction (Page 22)

Line 605 should be worded to clarify "a blunted sympathetic vascular transduction TO HIGH MSNA, WHICH WAS

OBSERVED IN BOTH SEXES, means that such sympathetic responses do not translate into an augmented pressor response". It is important to acknowledge blunted transduction was not seen with all analyses, and was not specific to women.

Referee #2:

-lines 265-276- This section would provide further clarity with reminders to the reader what responses they are considering.

"The "physiological sum" of the responses (i.e. ventilation or MSNA) to isocapnic hypoxia (peripheral chemoreflex activation) and hypercapnic hyperoxia (central chemoreflex activation) was calculated for comparison with the response to hypercapnic hypoxia (combined peripheral and central chemoreflex sensitivity)."

-lines 433-435- Typo here- should read while total MSNA remained unchanged in women.

"Jones et al. (1999) previously observed a ~20% reduction in total MSNA during 10 min hyperoxia (50% FiO₂) in young men, while total MSNA remained unchanged in men."

END OF COMMENTS

1st Confidential Review

02-Feb-2022

Once again, we are grateful to the Reviewing Editor, Senior Editor and both Referees for their careful consideration of our manuscript. As requested, we have provided a point-by-point response to the concerns raised (below) and both a clean version of our revised manuscript and a marked-up version showing the changes made in red underlined font (attached).

Reviewing Editor:

1. Thank you for re-submitting this manuscript. The manuscript is much improved. However, the readability is difficult in places and reviewer 1 has pointed out some places where this could be improved. There are also some inconsistencies within the manuscript which are pointed out by reviewer 1, such as inconsistencies in p-values. Please also check that exact p-values are reported: in the abstract there is a p-value reported as $P>0.05$.

Response: We are pleased that the manuscript is considered to be much improved. As indicated below, we have addressed all the points raised by Reviewer 1 and reported precise p-values throughout.

Please note that the $P>0.05$ previously included in the abstract referred to a collection of non-significant comparisons. This has not been replaced by 5 separate non-significant p-values (lines 59-60). Due to word restrictions, and the aforementioned readability issue, we have not been able to specify which refer to a main effect of sex (ANOVA) and which are a post hoc comparison. In the case of the latter two p-values are provided (total MSNA, VE) while in the latter former a single p-value is provided (MAP).

Senior Editor:

1. There is still a problem with reporting p values. The exact p values need to be reported in the tables.

Response: Our manuscript has been carefully reviewed and exact p-values are now stated throughout (including the tables).

2. *In the figures, please report the exact p values for trials. The only exception to this is if p is less than 0.0001, in which case '<' is permitted.*

Response: Exact p-values are now stated for trials in the figures.

3. *In the figure legends, please include: The exact p values for comparisons are shown in the statistical summary document.*

Response: The wording of this point makes it difficult to understand, and so we apologise if we are misinterpreting the point being made. We respectfully feel that adding p-values for post hoc comparisons to the figure legends is impractical. For example, Figure 3 alone contains 32 post hoc comparisons, 24 of which are significance at $p < 0.05$. We have included the rewritten Figure 3 legend with the information requested to illustrate our concern. We hope that including all the specific p-values referred to in the extensive statistical summary document (now ~100 pages in length) is satisfactory.

*“Figure 3. Minute ventilation ($\dot{V}E$) and muscle sympathetic nerve activity (MSNA; total activity) during eucapnia, isocapnic hypoxia, hypercapnic hyperoxia and hypercapnic hypoxia in women (red triangles, $n=10$) and men (blue circles, $n=10$). Panels A and C show absolute values. Panels B and D compare the change with the combined hypercapnic hypoxia trial versus the physiological sum of the responses to the separate isocapnic hypoxia and hypercapnic hyperoxia trials. The main effects of sex, breathing trial, and their interaction were examined using mixed model ANOVA analysis of variance (ANOVA) with repeated measures. Where a significant interaction is observed, differences identified during post hoc analysis (t-tests with Bonferroni correction) are identified as * $P < 0.05$ vs. women, † $P < 0.05$ vs. eucapnia, ‡ $P < 0.05$ vs. isocapnic hypoxia, § $P < 0.05$ vs. hypercapnic hyperoxia, # $P < 0.05$ vs. hypercapnic hypoxia. (For $\dot{V}E$ comparison within women, panel A: Isocapnic hypoxia vs. eucapnia: $P=1.000$; Hypercapnic hyperoxia vs. eucapnia: $P < 0.0001$; Hypercapnic hypoxia vs. eucapnia: $P < 0.0001$; Isocapnic hypoxia vs. hypercapnic hyperoxia: $P=0.0001$; Isocapnic hypoxia vs. hypercapnic hypoxia: $P < 0.0001$; Hypercapnic hyperoxia vs. hypercapnic hypoxia: $P=0.0001$. For $\dot{V}E$ comparison within men, panel A: Isocapnic hypoxia vs. eucapnia: $P=1.000$; Hypercapnic hyperoxia vs. eucapnia: $P < 0.0001$; Hypercapnic hypoxia*

vs. eucapnia: $P < 0.0001$; Isocapnic hypoxia vs. hypercapnic hyperoxia: $P < 0.0001$; Isocapnic hypoxia vs. hypercapnic hypoxia: $P < 0.0001$; Hypercapnic hyperoxia vs. hypercapnic hypoxia: $P = 0.0004$.

For $\dot{V}E$ comparison, panel B: Physiological sum of the $\dot{V}E$ responses to isocapnic hypoxia and hypercapnic hyperoxia vs. hypercapnic hypoxia: $P < 0.0001$. For total MSNA comparison within women, panel C: Isocapnic hypoxia vs. eucapnia: $P = 1.000$; Hypercapnic hyperoxia vs. eucapnia: $P < 0.0001$; Hypercapnic hypoxia vs. eucapnia: $P < 0.0001$; Isocapnic hypoxia vs. hypercapnic hyperoxia: $P = 0.002$; Isocapnic hypoxia vs. hypercapnic hypoxia: $P < 0.0001$; Hypercapnic hyperoxia vs. hypercapnic hypoxia: $P = 0.0003$. For total MSNA comparison within men, panel C: Isocapnic hypoxia vs. eucapnia: $P = 1.000$; Hypercapnic hyperoxia vs. eucapnia: $P = 0.01$; Hypercapnic hypoxia vs. eucapnia: $P = 0.003$; Isocapnic hypoxia vs. hypercapnic hyperoxia: $P = 0.02$; Isocapnic hypoxia vs. hypercapnic hypoxia: $P = 0.015$; Hypercapnic hyperoxia vs. hypercapnic hypoxia: $P = 0.03$. For total MSNA comparison, panel D: Physiological sum of the total MSNA responses to isocapnic hypoxia and hypercapnic hyperoxia vs. hypercapnic hypoxia: $P = 0.048$ ”

4. If the statistical summary document has errors please describe what is incorrect.

(Required):

The statistical summary document is not complete. All of the statistical comparisons shown in the figures need to be included in the document with the exact p values. This is particularly important since the authors chose to continue to use symbols in the figures.

Response: We have now included all the statistical comparisons shown in the figures in the statistical summary document.

5. In the results text, statistical comparisons not shown in tables or figures need to be included in the stat summary document.

Response: As advised, all statistical comparisons from the manuscript were included in the statistical summary document.

Referee #1:

1. The authors did a very nice job addressing prior comments. The comprehensiveness of the data collected will significantly advance knowledge within the field. In reviewing the revision, some minor issues should be addressed to ensure validity of conclusions and

readability. **Response:** Thank you for your supportive words. As indicated below, all the minor issues raised have been addressed.

2. The authors outline a number of post-hoc analyses by trial within the results section, however, these trial differences are not acknowledged within the tables. It would be helpful to indicate within the tables, using relevant symbols, results of the post-hoc trial comparisons.

Response: As suggested, symbols have been used to identify where a significant main effect of trial, but no interaction, was observed.

3. The authors should consider whether the terms men/women or male/female are more appropriate when describing their study populations. PMID: 34797173

Response: We have considered this and are satisfied that the terms men/women are appropriately used throughout the manuscript.

4. Figure 6 should include the statistical analysis used and the results, just as was done in the other figures.

Response: Figure 6 has been deleted. Statistical analysis was not undertaken on this data. It was used only to display temporal trends and magnitude of response and we now recognise that this is not necessary. The statistical analysis of this information was undertaken on data presented in Figure 5.

5. The authors state the BRS gain is reduced (Line 340). However, it appears the negative slope becomes more negative (supporting greater change in MSNA per unit change in BP). This reviewer interprets the results to show increased gain. This should be corrected throughout.

Response: Thank you for pointing out this typographical error. The revised Results (page 15, line 336) and Discussion (page 20, line 474) sections have been corrected to state that ABR-MSNA gain was increased during hypercapnic hyperoxia vs eucapnia.

6. In the methods Page 11, the authors describe BEI (baroreflex effectiveness index). This is not reported anywhere and should be removed.

Response: BEI data was included in Tables 2 and 5. The BEI response is now commented on in the revised Results section (page 15, lines 340-341 and page 16, line 372).

7. Within the abstract, the authors use $p=0.05$ to describe sex differences in the MSNA responses. It is unclear what this is referring to (MSNA amp? total MSNA?). These p-values could not be found within the body of the manuscript. The p-values/data reported in the abstract should be consistent with what is reported within the manuscript (perhaps total MSNA, $p=0.02$ and $p<0.001$; Lines 329-330).

Response: Thank you for pointing out this. Total MSNA is referred to in the Abstract. This information is now provided, and the highlighted p-values corrected (line 63-65). Please note that the p-values are all within the statistical summary document.

8. Line 307 states perception of breathlessness changed across trials, but was not different between men and women ($p=0.70$). The interaction value reported in Table 1 is $p=0.86$. What p-value are the authors referring to?

Response: The p-value ($p=0.70$, now stated as 0.699) on line 304 refers to the comparison between women and men for the perception of breathlessness (as shown in Table 1) and not the interaction p-value ($p=0.86$, now stated as 0.857).

9. *Figure 3D, which supports a trial effect (hyper-additive) is $p=0.05$. According to the results section, should this be $p=0.045$?*

Response: Thank you for pointing out this typographical error. This has been amended.

10. *This author appreciates acknowledging differences between hyperoxia trials with Jones et al (Lines 433-437). This may be more appropriately placed with discussion of method limitations where hyperoxia is similarly discussed (Page 23, Lines 554-561).*

Response: This information has been relocated as advised (page 23, lines 547-553).

11. *The authors acknowledge women were studied only one (Lines 571-574, Page 24). This may be more appropriately added to the other section discussing menstrual cycle (Line 586, Page 25)*

Response: This information has been relocated as advised (page 25, lines 583-585).

12. *Table 1: Men MSNA amplitude during hypercapnic hyperoxia is missing a symbol showing increase from eucapnia.*

Response: Thank you for pointing this out. The missing symbol has been added to Table 1.

13. *Line 483 and Line 484-485, "ABR-MSNA gain is unchanged during hypercapnic hypoxia" is stated twice.*

Response: This repeated statement has been deleted from the revised Discussion section.

14. *The authors may want to include new reference PMID: 34528146 when discussing hypoxia and transduction (Page 22)*

Response: This reference has been added as advised (page 21, line 501).

15. Line 605 should be worded to clarify "a blunted sympathetic vascular transduction TO HIGH MSNA, WHICH WAS OBSERVED IN BOTH SEXES, means that such sympathetic responses do not translate into an augmented pressor response". It is important to acknowledge blunted transduction was not seen with all analyses, and was not specific to women.

Response: This statement has been added to the perspective section as advised (page 25, line 596).

Referee #2:

1. Lines 265-276- This section would provide further clarity with reminders to the reader what responses they are considering. "The "physiological sum" of the responses (i.e. ventilation or MSNA) to isocapnic hypoxia (peripheral chemoreflex activation) and hypercapnic hyperoxia (central chemoreflex activation) was calculated for comparison with the response to hypercapnic hypoxia (combined peripheral and central chemoreflex sensitivity)."

Response: Thank you for the suggestion. The recommended wording has been adopted in the revised manuscript (page 12, lines 266-268)

3. Lines 433-435- Typo here- should read while total MSNA remained unchanged in women. "Jones et al. (1999) previously observed a ~20% reduction in total MSNA during 10 min hyperoxia (50% FiO2) in young men, while total MSNA remained unchanged in men."

Response: This statement has been corrected (page 23, lines 548-550).

Dear Dr Fisher,

Re: JP-RP-2022-282327R2 "Sex-differences in the sympathetic neurocirculatory responses to chemoreflex activation" by Ana Luiza C Sayegh, Jui-Lin Fan, Lauro C. Vianna, Mathew Dawes, Julian F. R. Paton, and James P Fisher

I am pleased to tell you that your paper has been accepted for publication in The Journal of Physiology.

NEW POLICY: In order to improve the transparency of its peer review process The Journal of Physiology publishes online as supporting information the peer review history of all articles accepted for publication. Readers will have access to decision letters, including all Editors' comments and referee reports, for each version of the manuscript and any author responses to peer review comments. Referees can decide whether or not they wish to be named on the peer review history document.

The last Word version of the paper submitted will be used by the Production Editors to prepare your proof. When this is ready you will receive an email containing a link to Wiley's Online Proofing System. The proof should be checked and corrected as quickly as possible.

Authors should note that it is too late at this point to offer corrections prior to proofing. The accepted version will be published online, ahead of the copy edited and typeset version being made available. Major corrections at proof stage, such as changes to figures, will be referred to the Reviewing Editor for approval before they can be incorporated. Only minor changes, such as to style and consistency, should be made a proof stage. Changes that need to be made after proof stage will usually require a formal correction notice.

All queries at proof stage should be sent to TJP@wiley.com

Are you on Twitter? Once your paper is online, why not share your achievement with your followers. Please tag The Journal (@jphysiol) in any tweets and we will share your accepted paper with our 23,000+ followers!

Yours sincerely,

Harold D Schultz
Senior Editor
The Journal of Physiology
<https://jp.msubmit.net>
<http://jp.physoc.org>
The Physiological Society
Hodgkin Huxley House
30 Farringdon Lane
London, EC1R 3AW
UK
<http://www.physoc.org>
<http://journals.physoc.org>

P.S. - You can help your research get the attention it deserves! Check out Wiley's free Promotion Guide for best-practice recommendations for promoting your work at www.wileyauthors.com/eeo/guide. And learn more about Wiley Editing Services which offers professional video, design, and writing services to create shareable video abstracts, infographics, conference posters, lay summaries, and research news stories for your research at www.wileyauthors.com/eeo/promotion.

*** IMPORTANT NOTICE ABOUT OPEN ACCESS ***

Information about Open Access policies can be found here <https://physoc.onlinelibrary.wiley.com/hub/access-policies>

To assist authors whose funding agencies mandate public access to published research findings sooner than 12 months after publication The Journal of Physiology allows authors to pay an open access (OA) fee to have their papers made freely available immediately on publication.

You will receive an email from Wiley with details on how to register or log-in to Wiley Authors Services where you will be able to place an OnlineOpen order.

You can check if your funder or institution has a Wiley Open Access Account here <https://authorservices.wiley.com/author-resources/Journal-Authors/licensing-and-open-access/open-access/author-compliance-tool.html>

Your article will be made Open Access upon publication, or as soon as payment is received.

If you wish to put your paper on an OA website such as PMC or UKPMC or your institutional repository within 12 months of

publication you must pay the open access fee, which covers the cost of publication.

OnlineOpen articles are deposited in PubMed Central (PMC) and PMC mirror sites. Authors of OnlineOpen articles are permitted to post the final, published PDF of their article on a website, institutional repository, or other free public server, immediately on publication.

Note to NIH-funded authors: The Journal of Physiology is published on PMC 12 months after publication, NIH-funded authors DO NOT NEED to pay to publish and DO NOT NEED to post their accepted papers on PMC.

EDITOR COMMENTS

Reviewing Editor:

Thank you for taking all comments on board. Well done on completing a complex integrative study.

Please provide an Author Contributions section when submitting your proof corrections.

REFEREE COMMENTS

Referee #1:

Thank you to the authors. I have no additional comments/concerns.

2nd Confidential Review

13-Apr-2022